# The Convex Relaxation Barrier, Revisited: Tightened Single-Neuron Relaxations for Neural Network Verification

**Christian Tjandraatmadja**
Google Research
ctjandra@google.com

**Ross Anderson**
Google Research
rander@google.com

**Joey Huchette**
Rice University
joehuchette@rice.edu

**Will Ma**
Columbia University
wm2428@gsb.columbia.edu

**Krunal Patel**
Polytechnique Montréal*
krunal.patel@polymtl.ca

**Juan Pablo Vielma**
Google Research
jvielma@google.com

## Abstract

We improve the effectiveness of propagation- and linear-optimization-based neural network verification algorithms with a new tightened convex relaxation for ReLU neurons. Unlike previous single-neuron relaxations which focus only on the univariate input space of the ReLU, our method considers the multivariate input space of the affine pre-activation function preceding the ReLU. Using results from submodularity and convex geometry, we derive an explicit description of the tightest possible convex relaxation when this multivariate input is over a box domain. We show that our convex relaxation is significantly stronger than the commonly used univariate-input relaxation which has been proposed as a natural convex relaxation barrier for verification. While our description of the relaxation may require an exponential number of inequalities, we show that they can be separated in linear time and hence can be efficiently incorporated into optimization algorithms on an as-needed basis. Based on this novel relaxation, we design two polynomial-time algorithms for neural network verification: a linear-programming-based algorithm that leverages the full power of our relaxation, and a fast propagation algorithm that generalizes existing approaches. In both cases, we show that for a modest increase in computational effort, our strengthened relaxation enables us to verify a significantly larger number of instances compared to similar algorithms.

## 1 Introduction

A fundamental problem in deep neural networks is to *verify* or *certify* that a trained network is *robust*, i.e. not susceptible to adversarial attacks [11, 29, 39]. Current approaches for neural network verification can be divided into *exact* (*complete*) methods and *relaxed* (*incomplete*) methods. Exact verifiers are often based on mixed integer programming (MIP) or more generally branch-and-bound [3, 4, 9, 10, 12, 14, 17, 24, 25, 31, 41, 48] or satisfiability modulo theories (SMT) [16, 19, 20, 27, 33] and, per their name, exactly solve the problem, with no false negatives or false positives. However, exact verifiers are typically based on solving NP-hard optimization problems [20] which can significantly limit their scalability. In contrast, relaxed verifiers are often based on polynomially-solvable optimization problems such as convex optimization or linear programming (LP) [2, 8, 15, 23, 26, 30, 32, 34, 47, 50], which in turn lend themselves to faster *propagation-based* methods where bounds are computed by a series of variable substitutions in a backwards pass through

the network [36, 44, 45, 46, 49]. Unfortunately, relaxed verifiers achieve this speed and scalability by trading off effectiveness (i.e. increased false negative rates), possibly failing to certify robustness when robustness is, in fact, present. As might be expected, the success of relaxed methods hinges on their tightness, or how closely they approximate the object which they are relaxing.

As producing the tightest possible relaxation for an entire neural network is no easier than the original verification problem, most relaxation approaches turn their attention instead to simpler substructures, such as individual neurons. For example, the commonly used $\Delta$-*relaxation*[2][16] is simple and offers the tightest possible relaxation for the univariate ReLU function, and as a result is the foundation for many relaxed verification methods. Recently, Salman et al. [32] characterized the *convex relaxation barrier*, showing that the effectiveness of all existing propagation-based fast verifiers is fundamentally limited by the tightness of this $\Delta$-relaxation. Unfortunately, they show computationally that this convex barrier can be a severe limitation on the effectiveness of relaxed verifiers based upon it. While the convex relaxation barrier can be bypassed in various ways (e.g. considering relaxations for multiple neurons [34]), as noted in [32, Appendix A] all existing approaches that achieve this do so by trading off clarity and speed.

In this paper we improve the effectiveness of propagation- and LP-based relaxed verifiers with a new tightened convex relaxation for ReLU neurons. Unlike the $\Delta$-relaxation which focuses only on the univariate input space of the ReLU, our relaxation considers the multivariate input space of the affine pre-activation function preceding the ReLU. By doing this, we are able to bypass the convex barrier from [32] while remaining in the realm of single-neuron relaxations that can be utilized by fast propagation- and LP-based verifiers.

More specifically, our contributions are as follows.

1. Using results from submodularity and convex geometry, we derive an explicit linear inequality description for the tightest possible convex relaxation of a single neuron, where, in the spirit of [3, 4], we take this to encompass the ReLU activation function, the affine pre-activation function preceding it, and known bounds on each input to this affine function. We show that this new convex relaxation is significantly stronger than the $\Delta$-relaxation, and hence bypasses the convex barrier from [32] without the need to consider multi-neuron interactions as in, e.g. [34].

2. We show that this description, while requiring an exponential number of inequalities in the worst case, admits an efficient separation routine. In particular, we present a linear time algorithm that, given a point, either asserts that this point lies within the relaxation, or returns an inequality that is not satisfied by this point. Using this routine, we develop two verification algorithms that incorporate our tighter inequalities into the relaxation.

   (a) `OptC2V`: We develop a polynomial-time LP-based algorithm that harnesses the full power of our new relaxation.

   (b) `FastC2V`: We develop a fast propagation-based algorithm that generalizes existing approaches (e.g. `Fast-Lin` [44] and `DeepPoly` [36]) by dynamically adapting the relaxation using our new inequalities.

3. Computational experiments on verification problems using networks from the ERAN dataset [38] demonstrate that leveraging these inequalities yields a substantial improvement in verification capability. In particular, our fast propagation-based algorithm surpasses the strongest possible algorithm restricted by the convex barrier (i.e. optimizing over the $\Delta$-relaxation at every neuron). We also show that our methods are competitive with more expensive state-of-the-art methods such as `RefineZono` [37] and `kPoly` [34], certifying more images than them in several cases.

## 2 Verification via mathematical optimization

Consider a neural network $f : \mathbb{R}^m \to \mathbb{R}^r$ described in terms of $N$ neurons in a linear order.[3] The first $m$ neurons are the *input neurons*, while the remaining *intermediate neurons* are indexed by

$i = m + 1, \ldots, N$. Given some input $x \in \mathbb{R}^m$, the relationship $f(x) = y$ can be described as

$$x_i = z_i \qquad\qquad\qquad \forall i = 1, \ldots, m \qquad\qquad\qquad \text{(the inputs)} \qquad \text{(1a)}$$

$$\hat{z}_i = \sum_{j=1}^{i-1} w_{i,j} z_j + b_i \quad \forall i = m+1, \ldots, N \qquad \text{(the pre-activation value)} \qquad \text{(1b)}$$

$$z_i = \sigma(\hat{z}_i) \qquad\qquad\quad \forall i = m+1, \ldots, N \qquad \text{(the post-activation value)} \qquad \text{(1c)}$$

$$y_i = \sum_{j=1}^{N} w_{i,j} z_j + b_i \quad \forall i = N+1, \ldots, N+r \qquad\qquad \text{(the outputs)}. \qquad \text{(1d)}$$

Here the constants $w$ and $b$ are the weights and biases, respectively, learned during training, while $\sigma(v) \stackrel{\text{def}}{=} \max\{0, v\}$ is the ReLU activation function. Appropriately, for each neuron $i$ we dub the variable $\hat{z}_i$ the *pre-activation variable* and $z_i$ the *post-activation variable*.

Given a trained network (i.e. fixed architecture, weights, and biases), we study a *verification problem* of the following form: given constant $c \in \mathbb{R}^r$, polyhedron $X \subseteq \mathbb{R}^m$, $\beta \in \mathbb{R}$, and

$$\gamma(c, X) \stackrel{\text{def}}{=} \max_{x \in X} c \cdot f(x) \equiv \max_{x,y,\hat{z},z} \left\{ c \cdot y \mid x \in X, \quad (1) \right\}, \qquad (2)$$

does $\gamma(c, X) \leqslant \beta$? Unfortunately, this problem is NP-hard [20]. Moreover, one is typically not content with solving just one problem of this form, but would like to query for many reasonable choices of $c$ and $X$ to be convinced that the network is robust to adversarial perturbations.

A promising approach to approximately solving the verification problem is to replace the intractable optimization problem defining $\gamma$ in (2) with a tractable *relaxation*. In particular, we aim to identify a tractable optimization problem whose optimal objective value $\gamma_R(c, X)$ satisfies $\gamma(c, X) \leqslant \gamma_R(c, X)$, for all parameters $c$ and $X$ of interest. Then, if $\gamma_R(c, X) \leqslant \beta$, we have answered the verification problem in the affirmative. However, note that it may well be the case that, by relaxing the problem, we may fail to verify a network that is, in fact, verifiable (i.e. $\gamma(c, X) \leqslant \beta < \gamma_R(c, X)$). Therefore, *the strength of our relaxation is crucial for reducing the false negative rate of our verification method.*

## 2.1 The $\Delta$-relaxation and its convex relaxation barrier

Salman et al. [32] note that many relaxation approaches for ReLU networks are based on the single-activation-function set $A^i \stackrel{\text{def}}{=} \{(\hat{z}_i, z_i) \in \mathbb{R}^2 \mid \hat{L}_i \leqslant \hat{z}_i \leqslant \hat{U}_i, \ z_i = \sigma_j(\hat{z}_i)\}$, where the pre-activation bounds $\hat{L}_i, \hat{U}_i \in \mathbb{R}$ are taken so that $\hat{L}_i \leqslant \hat{z}_i \leqslant \hat{U}_i$ for any point that satisfies $x \in X$ and (1). The $\Delta$-relaxation $C_\Delta^i \stackrel{\text{def}}{=} \text{Conv}(A^i)$ is optimal in the sense that it describes the convex hull of $A^i$, with three simple linear inequalities: $z_i \geqslant 0$, $z_i \geqslant \hat{z}_i$, and $z_i \leqslant \frac{\hat{U}_i}{\hat{U}_i - \hat{L}_i}(\hat{z}_i - \hat{L}_i)$.

The simplicity and small size of the $\Delta$-relaxation is appealing, as it leads to the relaxation

$$\gamma_\Delta(c, X) \stackrel{\text{def}}{=} \max_{x,y,\hat{z},z} \left\{ c \cdot y \mid x \in X, \quad (1a), (1b), (1d), \quad (\hat{z}_i, z_i) \in C_\Delta^i \ \forall i = m+1, \ldots, N \right\}. \quad (3)$$

This is a small[4] Linear Programming (LP) problem than is theoretically tractable and relatively easy to solve in practice. Moreover, a plethora of fast propagation-based algorithms [35, 36, 43, 44, 45, 49] center on an approach that can be interpreted as further relaxing $\gamma_\Delta$, where inequalities describing the sets $C_\Delta^i$ are judiciously dropped from the description in such a way that this LP becomes much easier to solve. Unfortunately, Salman et al. [32] observe that the quality of the verification bounds obtained through the $\Delta$-relaxation are intrinsically limited; a phenomenon they call the *convex relaxation barrier*. Nonetheless, this LP, along with faster propagation algorithms that utilize the inequalities defining $C_\Delta^i$, have been frequently applied to the verification task, often with substantial success.

## 2.2 Our approach: Eliding pre-activation variables

In this paper, we show that we can significantly improve over the accuracy of $\Delta$-relaxation verifiers with only a minimal trade-off in simplicity and speed. The key for this result is the observation that pre-activation variables are a "devil in disguise" in the context of convex relaxations. For a neuron $i$, the pre-activation variable $\hat{z}_i$ and the post-activation variable $z_i$ form the minimal set of variables needed to capture (and relax) the nonlinearity introduced by the ReLU. However, this

approach ignores the inputs to the pre-activation variable $\hat{z}_i$, i.e. the preceding post-activation variables $z_{1:i-1} \stackrel{\text{def}}{=} (z_1, \ldots, z_{i-1})$.

Our approach captures these relationships by instead turning our attention to the $i$-dimensional set[5] $S^i \stackrel{\text{def}}{=} \left\{ z \in \mathbb{R}^i \;\middle|\; L \leqslant z_{1:i-1} \leqslant U, \quad z_i = \sigma\left(\sum_{j=1}^{i-1} w_{i,j} z_j + b_i\right) \right\}$, where the post-activation bounds $L, U \in \mathbb{R}^{i-1}$ are such that $L_j \leqslant z_j \leqslant U_j$ for each point satisfying $x \in X$ and (1). Note that no pre-activation variables appear in this description; we elide them completely, substituting the affine function describing them inside of the activation function.

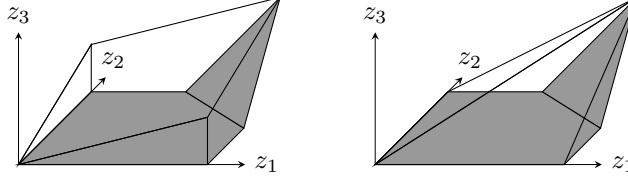

Figure 1: A simple neural network with $m = 2$ dimensional input and one intermediate neuron ($N = 3$). **(Left)** The feasible region for $\gamma_\Delta$, and **(Right)** The feasible region for $\gamma_{\texttt{Elide}}$. The $x$, $y$, and $\hat{z}$ variables, which depend affinely on the others, are projected out.

This immediately gives a single-neuron relaxation of the form

$$\gamma_{\texttt{Elide}}(c, X) \stackrel{\text{def}}{=} \max_{x,y,z} \left\{ c \cdot y \mid x \in X, \quad \text{(1a)}, \text{(1d)}, \quad z_{1:i} \in C^i_{\texttt{Elide}} \; \forall i = m+1, \ldots, N \right\}, \quad (4)$$

where $C^i_{\texttt{Elide}} \stackrel{\text{def}}{=} \text{Conv}(S^i)$ is the convex hull of $S^i$, as shown in Figure 1 (adapted from [3]), which contrasts it with the convex barrier and $\Delta$-relaxation. We will show that, unsurprisingly, $C^i_{\texttt{Elide}}$ will require exponentially many inequalities to describe in the worst case. However, *we show that this need not be a barrier to incorporating this tighter relaxation into verification algorithms.*

## 3  An exact convex relaxation for a single ReLU neuron

Let $w \in \mathbb{R}^n$, $b \in \mathbb{R}$, $f(x) \stackrel{\text{def}}{=} w \cdot x + b$, and $L, U \in \mathbb{R}^n$ be such that $L < U$. For ease of exposition, we rewrite the single-neuron set $S^i$ in the generic form

$$S \stackrel{\text{def}}{=} \{ (x, y) \in [L, U] \times \mathbb{R} \mid y = \sigma(f(x)) \}. \quad (5)$$

Notationally, take $[\![n]\!] \stackrel{\text{def}}{=} \{1, \ldots, n\}$, $\breve{L}_i \stackrel{\text{def}}{=} \begin{cases} L_i & w_i \geqslant 0 \\ U_i & \text{o.w.} \end{cases}$ and $\breve{U}_i \stackrel{\text{def}}{=} \begin{cases} U_i & w_i \geqslant 0 \\ L_i & \text{o.w.} \end{cases}$ for each $i \in [\![n]\!]$, $\ell(I) \stackrel{\text{def}}{=} \sum_{i \in I} w_i \breve{L}_i + \sum_{i \notin I} w_i \breve{U}_i + b$, and

$$\mathcal{J} \stackrel{\text{def}}{=} \left\{ (I, h) \in 2^{[\![n]\!]} \times [\![n]\!] \;\middle|\; \ell(I) \geqslant 0, \quad \ell(I \cup \{h\}) < 0, \quad w_i \neq 0 \; \forall i \in I \right\}.$$

Our main technical result uses results from submodularity and convex geometry [1, 6, 28, 40] to give the following closed-form characterization of $\text{Conv}(S)$. For a proof of Theorem 1, see Appendix A.

**Theorem 1.** *If $\ell([\![n]\!]) \geqslant 0$, then $\text{Conv}(S) = S = \{ (x, y) \in [L, U] \times \mathbb{R} \mid y = f(x) \}$. Alternatively, if $\ell(\varnothing) < 0$, then $\text{Conv}(S) = S = [L, U] \times \{0\}$. Otherwise, $\text{Conv}(S)$ is equal to the set of all $(x, y) \in \mathbb{R}^n \times \mathbb{R}$ satisfying*

$$y \geqslant w \cdot x + b, \quad y \geqslant 0, \quad L \leqslant x \leqslant U \quad (6a)$$

$$y \leqslant \sum_{i \in I} w_i (x_i - \breve{L}_i) + \frac{\ell(I)}{\breve{U}_h - \breve{L}_h}(x_h - \breve{L}_h) \quad \forall (I, h) \in \mathcal{J}. \quad (6b)$$

*Furthermore, if $d \stackrel{\text{def}}{=} |\{ i \in [\![n]\!] \mid w_i \neq 0 \}|$, then $d \leqslant |\mathcal{J}| \leqslant \lceil \frac{1}{2} d \rceil \binom{d}{\lceil \frac{1}{2} d \rceil}$ and for each of these inequalities (and each $d \in [\![n]\!]$) there exist data that makes it hold at equality.*

Note that this is the tightest possible relaxation when $x \in [L, U]$. Moreover, we observe that the relaxation offered by $\text{Conv}(S)$ can be arbitrarily tighter than that derived from the $\Delta$-relaxation.

**Proposition 1.** *For any input dimension $n$, there exists a point $\tilde{x} \in \mathbb{R}^n$, and a problem instance given by the affine function $f$, the $\Delta$-relaxation $C_\Delta$, and the single neuron set $S$ such that $\left(\max_{y:(f(\tilde{x}),y)\in C_\Delta} y\right) - \left(\max_{y:(\tilde{x},y)\in\text{Conv}(S)} y\right) = \Omega(n)$.*

Although the family of upper-bounding constraints (6b) may be exponentially large, the structure of the inequalities is remarkably simple. As a result, the *separation problem* can be solved efficiently: given $(x, y)$, either verify that $(x, y) \in \text{Conv}(S)$, or produce an inequality from the description (6) which is violated at $(x, y)$. For instance, we can solve in $\mathcal{O}(n \log n)$ time the optimization problem

$$v(x) \overset{\text{def}}{=} \min \left\{ \sum_{i\in I} w_i(x_i - \breve{L}_i) + \frac{\ell(I)}{\breve{U}_h - \breve{L}_h}(x_h - \breve{L}_h) \,\middle|\, (I, h) \in \mathcal{J} \right\}, \tag{7}$$

by sorting the indices with $w_i \neq 0$ in nondecreasing order of values $(x_i - \breve{L}_i)/(\breve{U}_i - \breve{L}_i)$, then adding them to $I$ in this order so long as $\ell(I) \geqslant 0$ (note that adding to $I$ can only decrease $\ell(I)$), and then letting $h$ be the index that triggered the stopping condition $\ell(I \cup \{h\}) < 0$. For more details, see the proof of Proposition 2 in Appendix B.

Then, to check if $(x, y) \in \text{Conv}(S)$, we first check if the point satisfies (6a), which can be accomplished in $\mathcal{O}(n)$ time. If so, we compute $v(x)$ in $\mathcal{O}(n \log n)$ time. If $y \leqslant v(x)$, then $(x, y) \in \text{Conv}(S)$. Otherwise, an optimal solution to (7) yields an inequality from (6b) that is most violated at $(x, y)$. In addition, we can also solve (7) slightly faster.

**Proposition 2.** *The optimization problem (7) can be solved in $\mathcal{O}(n)$ time.*

Together with the ellipsoid algorithm [18], Proposition 2 shows that the single-neuron relaxation $\gamma_{\texttt{Elide}}$ can be efficiently solved (at least in a theoretical sense).

**Corollary 1.** *If the weights $w$ and biases $b$ describing the neural network are rational, then the single-neuron relaxation (4) can be solved in polynomial time on the encoding sizes of $w$ and $b$.*

For proofs of Proposition 1, Proposition 2 and Corollary 1, see Appendix B.

**Connections with Anderson et al. [3, 4]** Anderson et al. [3, 4] have previously presented a MIP formulation that exactly models the set $S$ in (5). This formulation is *ideal* so, in particular, its LP relaxation offers a *lifted* LP formulation with one *auxiliary variable* whose projection onto the *original variables* $x$ and $y$ is exactly $\text{Conv}(S)$. Indeed, in Appendix A.2 we provide an alternative derivation for Theorem 1 using the machinery presented in [3]. This lifted LP can be used in lieu of our new formulation (6), though it offers no greater strength and requires an additional $N - m$ variables if applied for each neuron in the network. Moreover, it is not clear how to incorporate the lifted LP into propagation-based algorithms to be presented in the following section, which naturally work in the original variable space.

## 4 A propagation-based algorithm

We now present a technique to use the new family of strong inequalities (6b) to generate strong post-activation bounds for a trained neural network. A step-by-step example of this method is available in Appendix D. To properly define the algorithm, we begin by restating a generic propagation-based bound generation framework under which various algorithms from the literature are special cases (partially or completely) [35, 36, 43, 44, 45, 49].

### 4.1 A generic framework for computing post-activation bounds

Consider a bounded input domain $X \subseteq \mathbb{R}^m$, along with a single output (i.e. $r = 1$) to be maximized, which we name $\mathcal{C}(z) = \sum_{i=1}^\eta c_i z_i + b$ for some $\eta \leqslant N$. In this section, our goal is produce efficient algorithms for producing valid upper bounds for $\mathcal{C}$. First, let $z_i(x)$ denote the unique value of $z_i$ (post-activation variable $i$) implied by the equalities (1b–1c) when we set $z_{1:m} = x$ for some $x \in X$. Next, assume that for each intermediate neuron $i = m + 1, \ldots, \eta$ we have affine functions of the form $\mathcal{L}_i(z_{1:i-1}) = \sum_{j=1}^{i-1} w_{ij}^l z_j + b_i^l$ and $\mathcal{U}_i(z_{1:i-1}) = \sum_{j=1}^{i-1} w_{ij}^u z_j + b_i^u$, such that

$$\mathcal{L}_i(z_{1:i-1}(x)) \leqslant z_i(x) \leqslant \mathcal{U}_i(z_{1:i-1}(x)) \quad \forall x \in X, \quad i = 1, \ldots, \eta. \tag{8}$$

We consider how to construct these functions in the next subsection. Then, given these functions we can compute a bound on $\mathcal{C}\left(z_{1:\eta}\left(x\right)\right)$ through the following optimization problem:

$$B\left(\mathcal{C},\eta\right) \overset{\text{def}}{=} \max_{z} \quad \mathcal{C}(z) \equiv \sum_{i=1}^{\eta} c_i z_i + b \tag{9a}$$

$$\text{s.t.} \quad z_{1:m} \in X \tag{9b}$$

$$\mathcal{L}_i(z_{1:i-1}) \leqslant z_i \leqslant \mathcal{U}_i(z_{1:i-1}) \quad \forall i = m+1, \ldots, \eta. \tag{9c}$$

**Proposition 3.** *The optimal value of* (9) *is no less than* $\max_{x \in X} \mathcal{C}\left(z_{1:\eta}\left(x\right)\right)$.

The optimal value $B\left(\mathcal{C},\eta\right)$ can be quickly computed through propagation methods without explicitly computing an optimal solution to (9) [32, 49]. Such methods perform a backward pass to sequentially eliminate (project out) the intermediate variables $z_N, \ldots, z_{m+1}$, which can be interpreted as applying Fourier-Motzkin elimination [7, Chapter 2.8]. In a nutshell, for $i = \eta, \ldots, m+1$, the elimination step for variable $z_i$ uses its objective coefficient (which may be changing throughout the algorithm) to determine which one of the bounds from (9c) will be binding at the optimal solution and replaces $z_i$ by the expression $\mathcal{L}_i(z_{1:i-1})$ or $\mathcal{U}_i(z_{1:i-1})$ accordingly. The procedure ends with a smaller LP that only involves the input variables $z_{1:m}$ and can be quickly solved with an appropriate method. For instance, when $X$ is a box, as is common in verification problems, this final LP can be trivially solved by considering each variable individually. For more details, see Algorithm 1 in Appendix C.

## 4.2 Selecting the bounding functions

The framework described in the previous section required as input the family of bounding functions $\{\mathcal{L}_i, \mathcal{U}_i\}_{i=m+1}^{\eta}$. A typical approach to generate these will proceed sequentially, deriving the $i$-th pair of functions using *scalar bounds* $\hat{L}_i, \hat{U}_i \in \mathbb{R}$ on the $i$-th pre-activation variables $\hat{z}_i$, which by (1b) is equal to $\sum_{j=1}^{i-1} w_{i,j} z_j + b_i$. Hence, these scalar bounds must satisfy

$$\hat{L}_i \leqslant \sum_{j=1}^{i-1} w_{i,j} z_j(x) + b_i \leqslant \hat{U}_i \quad \forall x \in X. \tag{10}$$

These bounds can then be used as a basis to linearize the nonlinear equation

$$z_i = \sigma\left(\sum_{j=1}^{i-1} w_{i,j} z_j + b_i\right) \tag{11}$$

implied by (1b-1c). If $\hat{U}_i \leqslant 0$ or $\hat{L}_i \geqslant 0$, then (11) behaves linearly when (10) holds, and so we can let $\mathcal{L}_i(z_{1:i-1}) = \mathcal{U}_i(z_{1:i-1}) = \sum_{j=1}^{i-1} w_{i,j} z_j + b_i$ or $\mathcal{L}_i(z_{1:i-1}) = \mathcal{U}_i(z_{1:i-1}) = 0$, respectively. Otherwise, we can construct non-trivial bounds such as

$$\mathcal{L}_i(z_{1:i-1}) = \frac{\hat{U}_i}{\hat{U}_i - \hat{L}_i}\left(\sum_{j=1}^{i-1} w_{i,j} z_j + b_i\right) \quad \text{and} \quad \mathcal{U}_i(z_{1:i-1}) = \frac{\hat{U}_i}{\hat{U}_i - \hat{L}_i}\left(\sum_{j=1}^{i-1} w_{i,j} z_j + b_i - \hat{L}_i\right),$$

which can be derived from the $\Delta$-relaxation: $\mathcal{U}_i(z_{1:i-1})$ is the single upper-bounding inequality present on the left side of Figure 1, and $\mathcal{L}_i(z_{1:i-1})$ is a shifted down version of this inequality.[6] This pair is used by algorithms such as Fast-Lin [44], DeepZ [35], Neurify [43], and that of Wong and Kolter [45]. Algorithms such as DeepPoly [36] and CROWN-Ada [49] can be derived by selecting the same $\mathcal{U}_i(z_{1:i-1})$ as above and $\mathcal{L}_i(z_{1:i-1}) = 0$ if $|\hat{L}_i| \geqslant |\hat{U}_i|$ or $\mathcal{L}_i(z_{1:i-1}) = \sum_{j=1}^{i-1} w_{i,j} z_j + b_i$ otherwise (i.e. whichever yields the smallest area of the relaxation). In the next subsection, we propose using (6b) for $\mathcal{U}_i(z_{1:i-1})$.

Scalar bounds satisfying (10) for the $i$-th pre-activation variable can be computed by letting $\mathcal{C}^{U,i}\left(z_{1:(i-1)}\right) = \sum_{j=1}^{i-1} w_{i,j} z_j + b_i$ and then setting $\hat{L}_i = -B\left(\mathcal{C}^{L,i}, i-1\right)$ and $\hat{U}_i = B\left(\mathcal{C}^{U,i}, i-1\right)$. Therefore, to reach a final bound for $\eta = N$, we can iteratively compute $\hat{L}_i$ and $\hat{U}_i$ for $i = m+1, \ldots, N$ by solving (9) each time, since each of these problems requires only affine bounding functions up to intermediate neuron $i-1$. See Algorithm 4 in Appendix C for details.

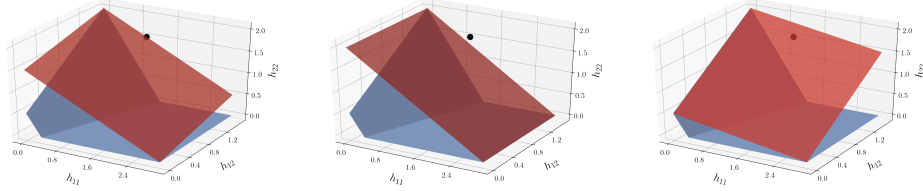

(a) An inequality from the $\Delta$-relaxation.    (b) An inequality from (6b).    (c) Another inequality from (6b).

Figure 2: Possible choices of upper bounding functions $\mathcal{U}_i$ for a single ReLU. The black point depicts a solution $z_{1:N}$ that we would like to separate (projected to the input-output space of the ReLU), which is cut off by the inequality in (b). A full example involving these particular inequalities can be found in Appendix D.

### 4.3 Our contribution: Tighter bounds by dynamically updating bounding functions

In Theorem 1 we have derived a family of inequalities, (6b), which can be applied to yield valid upper bounding affine functions for each intermediate neuron in a network. As there may be exponentially many such inequalities, it is not clear *a priori* which to select as input to the algorithm from Section 4.1. Therefore, we present a simple iterative scheme in which we apply a small number of solves of (9), incrementally updating the set of affine bounding functions used at each iteration.

Our goal is to update the upper bounding function $\mathcal{U}_i$ with one of the inequalities from (6b) as illustrated in Figure 2 via the separation procedure of Proposition 2, which requires an optimal solution $z_{1:N}$ for (9). However, the backward pass of the propagation algorithm described in Section 4.1 only computes the optimal value $B\left(\mathcal{C}, \eta\right)$ and a partial solution $z_{1:m}$. For this reason, we first extend the propagation algorithm with a forward pass that completes the partial solution $z_{1:m}$ by propagating the values for $z_{m+1}, \ldots, z_N$ through the network. This propagation uses the same affine bounding functions from (9c) that were used to eliminate variables in the backward pass. For more details, see Algorithm 2 in Appendix C.

In essence, our complete dynamic algorithm initializes with a set of bounding functions (e.g. from `Fast-Lin` or `DeepPoly`), applies a backward pass to solve the bounding problem, and then a forward pass to reconstruct the full solution. It then takes that full solution, and at each intermediate neuron $i$ applies the separation procedure of Proposition 2 to produce an inequality from the family (6b). If this inequality is violated, it replaces the upper bounding function $\mathcal{U}_i$ with this inequality from (6b). We repeat for as many iterations as desired and take the best bound produced across all iterations. In this way, we use separation to help us select from a large family just one inequality that will (hopefully) be most beneficial for improving the bound. For more details, see Algorithm 3 in Appendix C.

## 5 Computational experiments

### 5.1 Computational setup

We evaluate two methods: the propagation-based algorithm from Section 4.3 and a method based on partially solving the LP from Theorem 1 by treating the inequalities (6b) as cutting planes, i.e. inequalities that are dynamically added to tighten a relaxation. To focus on the benefit of incorporating the inequalities (6b) into verification algorithms, we implement simple versions of the algorithms, devoid of extraneous features and fine-tuning. We name this framework "Cut-to-Verify" (C2V), and the propagation-based and LP-based algorithms `FastC2V` and `OptC2V`, respectively. See `https://github.com/google-research/tf-opt` for the implementation.

The overall framework in both methods is the same: we compute scalar bounds for the pre-activation variables of all neurons as we move forward in the network, using those bounds to produce the subsequent affine bounding functions and LP formulations as discussed in Section 4.2. Below, we describe the bounds computation for each individual neuron.

**Propagation-based algorithm (`FastC2V`).**  We implement the algorithm described in Section 4.3, using the initial affine bounding functions $\{\mathcal{L}_i, \mathcal{U}_i\}_{i=m+1}^{N}$ from `DeepPoly` [36] and `CROWN-Ada` [49], as described in Section 4.1.[7] In this implementation, we run a single iteration of the algorithm.

**LP-based algorithm (`OptC2V`).**  Each bound is generated by solving a series of LPs where our upper bounding inequalities are dynamically generated and added as cutting planes. We start with the standard $\Delta$-relaxation LP, solve it to optimality, and then for every neuron preceding the one we are bounding, we add the most violated inequality with respect to the LP optimum by solving (7). This can be repeated multiple times. In this implementation, we perform three rounds of separation. We generate new cuts from scratch for each bound that we compute.

In both methods, at each neuron we take the best between the bound produced by the method and the trivial interval arithmetic bound. Appendix E contains other implementation details.

We compare each of our novel algorithms against their natural baselines: `DeepPoly` for our propagation-based method, and the standard $\Delta$-relaxation LP for our cutting plane method. Our implementation of `DeepPoly` is slightly different from the one in [36] in that we take the best of interval arithmetic and the result of `DeepPoly` at each neuron. Moreover, our implementation is sequential, even though operations in the same layer could be parallelized (for each of the algorithms implemented in this work). The LP method simply solves the $\Delta$-relaxation LP to generate bounds at each neuron. In addition, we compare them with `RefineZono` [37] and `kPoly` [34], two state-of-the-art incomplete verification methods.

**Verification problem.**  We consider the following verification problem: given a correctly labeled target image, certify that the neural network returns the same label for each input within $L_\infty$-distance at most $\epsilon$ of that target image. More precisely, given an image $\hat{x} \in [0,1]^m$ correctly labeled as $t$, a neural network where $f_k(x)$ returns its logit for class $k \in K$, and a distance $\epsilon > 0$, the image $\hat{x}$ is verified to be robust if $\max_{x \in [\hat{L}, \hat{U}]} \max_{k \in K} \{f_k(x) - f_t(x)\} < 0$, where $\hat{L}_i = \max\{0, \hat{x}_i - \epsilon\}$ and $\hat{U}_i = \min\{1, \hat{x}_i + \epsilon\}$ for all $i = 1, \ldots, m$. For propagation-based methods, the inner $\max$ term can be handled by computing bounds for $f_k(x) - f_t(x)$ for every class $k \neq t$ and checking if the maximum bound is negative, although we only need to compute pre-activation bounds throughout the network once. For LP-based methods, this inner term can be incorporated directly into the model.

To facilitate the comparison with existing algorithms, our experimental setup closely follows that of Singh et al. [34]. We experiment on a subset of trained neural networks from the publicly available ERAN dataset [38]. We examine the following networks: the fully connected ReLU networks 6x100 ($\epsilon = 0.026$), 9x100 ($\epsilon = 0.026$), 6x200 ($\epsilon = 0.015$), 9x200 ($\epsilon = 0.015$), all trained on MNIST without adversarial training; the ReLU convolutional networks ConvSmall for MNIST ($\epsilon = 0.12$), with 3 layers and trained without adversarial training; the ReLU network ConvBig for MNIST ($\epsilon = 0.3$), with 6 layers and trained with DiffAI; and the ReLU network ConvSmall for CIFAR-10 ($\epsilon = 2/255$), with 3 layers and trained with PGD. These $\epsilon$ values are the ones used in [34] and they are cited as being challenging. For more details on these networks, see Appendix E or [38]. For each network, we verify the first 1000 images from their respective test sets except those that are incorrectly classified.

Due to numerical issues with LPs, we zero out small values in the convolutional networks for the LP-based algorithms (see Appendix E). Other than this, we do not perform any tuning according to instance. Our implementation is in C++ and we perform our experiments in an Intel Xeon E5-2699 2.3Ghz machine with 128GB of RAM. We use Gurobi 8.1 as the LP solver, take advantage of incremental solves, and set the LP algorithm to dual simplex, as we find it to be faster for these LPs in practice. This means that our LP implementation does not run in polynomial time, even though it could in theory by using a different LP algorithm (see Corollary 1).

To contextualize the results, we include an upper bound on the number of verifiable images. This is computed with a standard implementation of gradient descent with learning rate 0.01 and 20 steps. For each image, we take 100 random initializations (10 for MNIST ConvBig and CIFAR-10 ConvSmall) and check if the adversarial example produced by gradient descent is valid. The upper bound is the number of images for which we were unable to produce an adversarial example.

Table 1: Number of images verified and average verification times per image for a set of networks from the ERAN dataset [38]. ConvS and ConvB denote ConvSmall and ConvBig, respectively. Results for `RefineZono` and `kPoly` are taken from [34].

| Method | | MNIST | | | | | | CIFAR-10 |
|---|---|---|---|---|---|---|---|---|
| | | 6x100 | 9x100 | 6x200 | 9x200 | ConvS | ConvB | ConvS |
| `DeepPoly` | #verified | 160 | 182 | 292 | 259 | 162 | 652 | 359 |
| | Time (s) | 0.7 | 1.4 | 2.4 | 5.6 | 0.9 | 7.4 | 2.8 |
| `FastC2V` | #verified | 279 | 269 | 477 | 392 | 274 | 691 | 390 |
| | Time (s) | 8.7 | 19.3 | 25.2 | 57.2 | 5.3 | 16.3 | 15.3 |
| `LP` | #verified | 201 | 223 | 344 | 307 | 242 | 743 | 373 |
| | Time (s) | 50.5 | 385.6 | 218.2 | 2824.7 | 23.1 | 24.9 | 38.1 |
| `OptC2V` | #verified | 429 | 384 | 601 | 528 | 436 | 771 | 398 |
| | Time (s) | 136.7 | 759.4 | 402.8 | 3450.7 | 55.4 | 102.0 | 104.8 |
| `RefineZono` | #verified | 312 | 304 | 341 | 316 | 179 | 648 | 347 |
| `kPoly` | #verified | 441 | 369 | 574 | 506 | 347 | 736 | 399 |
| Upper bound | #verified | 842 | 820 | 901 | 911 | 746 | 831 | 482 |

## 5.2 Computational results

The computational results in Table 1 demonstrate that adding the upper bounding inequalities proposed in this paper significantly improves the number of images verified compared to their base counterparts. While on average `FastC2V` spends an order of magnitude more time than `DeepPoly` to achieve this, it still takes below one minute on average for all instances examined. `OptC2V` takes approximately 1.2 to 2.7 times of a pure LP method to generate bounds in the problems examined. Since we start from the LP basis of the previous solve, subsequent LPs after adding cuts are generally faster.

Interestingly, we observe that `FastC2V` verifies more images than LP in almost all cases in much less time. This indicates that, in practice, a two-inequality relaxation with a single (carefully chosen) tighter inequality from (6b) can often be stronger than the three-inequality $\Delta$-relaxation.

When compared to other state-of-the-art incomplete verifiers, we observe that for the larger networks, improving `DeepPoly` with our inequalities enables it to verify more images than `RefineZono` [37], a highly fine-tuned method that combines MIP, LP, and `DeepPoly`, but without the expensive computation and the parameter tuning needs from `RefineZono`. In addition, we find that adding our inequalities to LPs is competitive with `kPoly`, surpassing it for some of the networks. While the timings in [34] may not be comparable to our timings, the authors report average times for `RefineZono` and `kPoly` within the range of 4 to 15 minutes and 40 seconds to 8 minutes, respectively.

Appendix F contains additional computational results where we consider multiple trained networks and distances $\epsilon$ from the base image.

**Outlook: Our methods as subroutines** The scope of our computational experiments is to demonstrate the practicality and strength of our full-neuron relaxation applied to simple methods, rather than to engineer full-blown state-of-the-art verification methods. Towards such a goal, we remark that both `RefineZono` and `kPoly` rely on LP and other faster verification methods as building blocks to a stronger method, and either of our methods could be plugged into them. For example, we could consider a hybrid approach similar to `RefineZono` that uses the stronger, but slower `OptC2V` in the earlier layers (where it can have the most impact) and then switches to `FastC2V`, which could result in verification times closer to `FastC2V` with an effectiveness closer to `OptC2V`. In addition, `kPoly` exploits the correlation between multiple neurons in the same layer, whereas our approach does not, suggesting that there is room to combine approaches. Finally, we note that solving time can be controlled with a more careful management of the inequalities to be added and parallelizing bound computation of neurons in the same layer.

## Broader Impact

In a world where deep learning is impacting our lives in ever more tangible ways, verification is an essential task to ensure that these black box systems behave as we expect them to. Our fast, simple algorithms have the potential to make a positive impact by verifying a larger number of inputs to be robust within a short time-frame, often required in several applications. Of course, we should be cautious that although our algorithms provide a mathematical certificate of an instance being robust, failure to use the system correctly, such as modeling the verification problem in a way that does not reflect real-world concerns, can still lead to unreliable neural networks. We also highlight that our version of the verification problem, while accurately capturing a reasonable formal specification of robustness, clearly does not perfectly coincide with "robustness" as may be used in a colloquial sense. Therefore, we highlight the importance of understanding the strengths and limitations of the mathematical model of verification used, so that a false sense of complacency does not set in.

## Funding Disclosure

No third-party funding was received for this work.

## Footnotes

*This work was completed while this author was at Google Research.

[2]Sometimes also called the *triangle relaxation* [22, 34].

[3]This allows us to consider feedforward networks, including those that skip layers (e.g. see [32, 50]).

[4]Here, "small" means the number of variables and constraints is $\mathcal{O}(\# \text{ of neurons})$.

[5]The *effective* dimension of this set can be much smaller if $w_{i,\cdot}$ is sparse. This is the case with a feedforward network, where the number of nonzeros is (at most) the number of neurons in the preceding layer.

[6]Note that these functions satisfy (8) only when $\hat{U}_i > 0$ and $\hat{L}_i < 0$.

[7]Our framework supports initializing from the `Fast-Lin` inequalities as well, but it has been observed that the inequalities from `DeepPoly` perform better computationally.

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
