[Supplementary Material]

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

[8]See also [5, Theorem 7.9]: the proof of [28, Lemma 2] can be readily adapted to accommodate non-strict, rather than strict, linear inequalities.

[9] Such an affine interpolation exists and is unique because $\mathbf{V}_\pi$ is a set of $n+1$ affinely independent points.

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

# A  Proof of Theorem 1

We provide two different proofs for Theorem 1. The first proof is based on classical machinery from submodular and convex optimization. The alternative proof is based on projecting down an extended MIP formulation built using disjunctive programming. We include them both since each proof provides unique insights on our new relaxation.

We first state a lemma that is used by both proofs for bounding the number of inequalities. Notationally, we will take $\mathbf{0}^d$ and $\mathbf{1}^d$ as the length $d$ vectors of all zeros and all ones, respectively, and $e(i) \in \mathbb{R}^n$ for $i \in [\![n]\!]$ as the $i$-th canonical unit vector, where the length will be implicitly determined by the context of its use. In some cases it will be convenient to refer to the 0-th canonical vector $e(0) = \mathbf{0}^n$.

**Lemma 1.** *If $u, v \in \{0,1\}^d$ are such that $\sum_{i=1}^{d} |u_i - v_i| = 1$, then we say that $uv$ is an edge of $[0,1]^d$. For $w \in \mathbb{R}^d$ and $b \in \mathbb{R}$, we say the hyperplane $w \cdot x + b = 0$ cuts edge $uv$ of $[0,1]^d$ if $w \cdot u + b < 0$ and $w \cdot v + b \geq 0$. If $b < 0$ and $\sum_{i=1}^{d} w_i + b \geq 0$, then the number of edges cut by one such hyperplane is lower-bounded by $d$ and upper-bounded by $\lceil \frac{1}{2} d \rceil \binom{d}{\lceil \frac{1}{2} d \rceil}$. For each bound there exists a hyperplane with $w \in \mathbb{R}_+^d$ such that the bounds holds at equality.*

*Proof.* Consider the graph $G = (V, E)$ with $V = \{0,1\}^d$ and $E$ equal to the edges of $[0,1]^d$. Let $s = \mathbf{0}^d$ and $t = \mathbf{1}^d$. Then $w \cdot s + b < 0$ and $w \cdot t + b \geq 0$, so the edges of $[0,1]^d$ cut by the hyperplane form a $s - t$ graph-cut in $G$ (note that this does *not* have the same meaning as the definition of cut for an edge given in the Lemma statement). Hence, the number of edges cut by the hyperplane are lower bounded by $d$ (e.g. follows by Menger's theorem by noting that there are $d$ disjoint paths in $G$ from $s$ to $t$). An example of a hyperplane that achieves this lower bound is $w = \mathbf{1}^d$ and $b = -1/2$.

The tight upper bound follows from a simple adaptation of the proof of a result from [28].[8] An example of a hyperplane that achieves this upper bound is $w = \mathbf{1}^d$ and $b = -\lceil \frac{1}{2} d \rceil$. $\square$

## A.1  A proof using submodularity

We start with an example.

### A.1.1  Illustrative example and definitions

**Example 1.** *Consider the set from* (5) *for $n = 2$, $w = (1,1)$, $b = (-1.5)$, $L = (0,0)$ and $U = (0,0)$, which corresponds to*

$$S = \left\{ (x,y) \in [0,1]^2 \times \mathbb{R} \mid y = g(x) \right\}$$

*for $g(x) \overset{\text{def}}{=} \max\left\{ 0, x_1 + x_2 - 1.5 \right\}$. Set $S$ is depicted in Figure 3 and we can check that $\mathrm{Conv}(S)$ is described by*

$$x \in [0,1]^2 \tag{12a}$$

$$y \geq g(x) \tag{12b}$$

$$y \leq r_1(x), \quad y \leq r_2(x) \tag{12c}$$

*for $r_1(x) \overset{\text{def}}{=} 0.5 x_2$ and $r_2(x) \overset{\text{def}}{=} 0.5 x_1$. Inequality* (12b) *is obtained by relaxing the equation describing $S$ to an inequality and using the fact that $g(x)$ is convex. Functions $r_1$ and $r_2$ from inequality* (12c) *are depicted in Figures 3a and 3b, respectively. These functions can be obtained through the following interpolation procedure.*

*First, consider the subdivision of $[0,1]^2$ into the triangles $\mathbf{T}_1$ and $\mathbf{T}_2$ depicted in Figures 3a and 3b, respectively. As depicted Figure 3a, the vertices of $\mathbf{T}_1$ are obtained by incrementally adding the canonical vectors to $(0,0)$, in order, until we obtain $(1,1)$. That is, the vertices of $\mathbf{T}_1$ are $e(0) = (0,0)$, $e(0) + e(1) = e(1) = (0,1)$ and $e(0) + e(1) + e(2) = (1,1)$. In contrast, as depicted in Figure 3b, the vertices of $\mathbf{T}_2$ are obtained by incrementally adding the canonical vectors in reverse order (i.e. the vertices of $\mathbf{T}_2$ are $e(0) = (0,0)$, $e(0) + e(2) = (1,0)$ and $e(0) + e(2) + e(1) = (1,1)$).*

*Second, we obtain $r_1$ and $r_2$ by constructing the unique affine interpolation of $g$ on $\mathbf{T}_1$ and $\mathbf{T}_2$, respectively. That is, as depicted in Figure 3a, $r_1(x) = \alpha^1 \cdot x + \beta_1$, where $\alpha^1 \in \mathbb{R}^2$ and $\beta_1 \in \mathbb{R}$ are such that $r_1$ is equal to $g$ for the three vertices $(0,0)$, $(1,0)$ and $(1,1)$ of $\mathbf{T}_1$:*

$$\begin{pmatrix} 0 & 0 \\ 1 & 0 \\ 1 & 1 \end{pmatrix} \alpha^1 + \beta_1 = \begin{pmatrix} g(0,0) \\ g(1,0) \\ g(1,1) \end{pmatrix} = \begin{pmatrix} 0 \\ 0 \\ 0.5 \end{pmatrix}.$$

*The unique solution of this system is $\alpha^1 = (0, 0.5)$ and $\beta_1 = 0$, which yields $r_1(x) = 0.5x_2$. Function $r_2$ is obtained by a similar procedure using the vertices of $\mathbf{T}_2$ as illustrated Figure 3b.*

(a) Constructing $r_1$ using $\mathbf{T}_1$.    (b) Constructing $r_2$ using $\mathbf{T}_2$.    (c) Checking membership in $\mathrm{Conv}(S)$.

Figure 3: Using interpolation on triangles to construct $\mathrm{Conv}(S)$ for Example 1.

The subdivision of $[0,1]^2$ into $\mathbf{T}_1$ and $\mathbf{T}_2$ can be extended to $[0,1]^n$ by considering all $n!$ possible orders in which we can obtain $\mathbf{1}^n$ from $\mathbf{0}^n$ by incrementally adding the canonical vectors. We represent these orders using the set of all permutations of $[\![n]\!]$. In Example 1, this set is given by $\mathcal{S}_2 \overset{\text{def}}{=} \{\pi_1, \pi_2\}$, where $\pi_i : [\![2]\!] \to [\![2]\!]$ for each $i \in [\![2]\!]$, $\pi_1(1) = 1$, $\pi_1(2) = 2$, $\pi_2(1) = 2$, and $\pi_2(2) = 1$. Then, under the notation of Definition 1 below, we have $\mathbf{T}_1 = \mathbf{T}_{\pi_1}$ and $\mathbf{T}_2 = \mathbf{T}_{\pi_2}$.

**Definition 1.** *Let $\mathcal{S}_n$ be the set of all permutations of $[\![n]\!]$. Then for every $\pi \in \mathcal{S}_n$, we define $\mathbf{V}_\pi = \left\{ \sum_{i=0}^{j} e\left(\pi\left(i\right)\right) \right\}_{j=0}^{n}$ and*

$$\mathbf{T}_\pi = \mathrm{conv}\left(\mathbf{V}_\pi\right) = \left\{ x \in \mathbb{R}^n \mid 1 \geqslant x_{\pi(1)} \geqslant x_{\pi(2)} \geqslant \ldots \geqslant x_{\pi(n)} \geqslant 0 \right\}. \tag{13}$$

The collection of simplices $\{\mathbf{T}_\pi\}_{\pi \in \mathcal{S}_n}$, whose union is $[0,1]^n$, is known as the *Kuhn triangulation* of $[0,1]^n$ [42].

The number of simplices in the Kuhn triangulation is exponential, so an $n$-dimensional generalization of Example 1 could contain an exponential number of inequalities in (12c). Fortunately, as illustrated in the following example, the characterization of $\mathbf{T}_\pi$ in the right hand side of (13) allow us to easily filter for relevant inequalities.

**Example 1 continued.** *Consider the point $(x^*, y^*) = (0.6, 0.3, 0.5)$ depicted as a red star in Figure 3c. To check if $(x^*, y^*) \in \mathrm{Conv}(S)$ we can first verify that $y^* \geqslant g(x^*)$ and $x^* \in [0,1]^2$. It then only remains to check that $(x^*, y^*)$ satisfies all inequalities in (12c). However, we can instead exploit the fact that if $x \in \mathbf{T}_1$, then $r_1(x) = \min\{r_1(x), r_2(x)\}$. As illustrated in Figure 3c we can use the fact that $x_1^* \geqslant x_2^*$ to conclude that $x^*$ (depicted as a red circle in Figure 3c) belongs to $\mathbf{T}_1$. Finally, we can check that $r_1(x^*) = 0.3 < 0.5$ to conclude that $(x^*, y^*) \notin \mathrm{Conv}(S)$ (Point $(x^*, 0.3)$ is depicted as a red diamond in Figure 3c).*

To show that the ideas in Example 1 can be generalized, we will exploit properties of submodular functions. For that we connect functions from $[0,1]^n$ with set-functions. We pick one specific connection that simplifies the statement and proof of Theorem 1.

**Definition 2.** *A set-function $H : 2^{[\![n]\!]} \to \mathbb{R}$ is submodular if*

$$H(S) + H(T) \geqslant H(S \cup T) + H(S \cap T) \quad \forall S, T \subseteq [\![n]\!].$$

*For any $h : [0,1]^n \to \mathbb{R}$ we define the set-function $H : 2^{[\![n]\!]} \to \mathbb{R}$ given by $H(I) = h\left(\sum_{i \notin I} e(i)\right)$ for each $I \subseteq [\![n]\!]$. In particular, $H([\![n]\!]) = h(\mathbf{0}^n)$ and $H(\varnothing) = h(\mathbf{1}^n)$. In general, for any function from $[0,1]^n$ to $\mathbb{R}$ defined as a lower case letter (e.g. $h$), we let the associated set-function be defined by the upper case version of this letter (e.g. $H$).*

### A.1.2    Proof of Theorem 1

Our proof has three steps. First, we formalize the idea in Example 1 for arbitrary dimensions (Theorem 2). Then, we reduce the number of inequalities by characterizing which of the simplices $\mathbf{T}_\pi$ lead to identical inequalities (Lemma 2). Finally, to complete the proof of Theorem 1, we describe the explicit form of these inequalities.

Corollary 3.14 in [40] gives us a precise description of $\mathrm{Conv}(Q)$ where $Q$ is a normalized version of $S$ from (5) that also considers any convex activation function. We include a submodularity-based proof of the corollary for completeness, adapted to our context.

**Theorem 2.** *Let $w \in \mathbb{R}_+^n$ and $b \in \mathbb{R}$, $f(x) = w \cdot x + b$, $\rho : \mathbb{R} \to \mathbb{R}$ be any convex function, $g(x) = \rho(f(x))$ and $Q = \{(x,y) \in [0,1]^n \times \mathbb{R} \mid y = \rho(f(x))\}$.*

*For each $\pi \in \mathcal{S}_n$ let $r_\pi : [0,1]^n \to \mathbb{R}$ be the unique affine interpolation[9] of $g$ on $\mathbf{T}_\pi$ such that $r_\pi(v) = g(v)$ for all $v \in \mathbf{V}_\pi$. Then $\mathrm{Conv}(Q)$ equals the set of all $(x,y) \in \mathbb{R}^n \times \mathbb{R}$ satisfying*

$$y \geqslant g(x) \tag{14a}$$
$$y \leqslant r_\pi(x) \qquad\qquad \forall \pi \in \mathcal{S}_n \tag{14b}$$
$$0 \leqslant x_i \leqslant 1 \qquad\qquad \forall i \in [\![n]\!] \tag{14c}$$

*Proof.* Let $h : [0,1]^n \to \mathbb{R}$ be such that $h(x) = -g(x) = -\rho(f(x))$ for all $x \in [0,1]^n$. In addition, let $\underline{h}$ and $\overline{h}$ respectively be the convex and concave envelopes of $h$ (i.e. the largest convex underestimator of $h$, which is well-defined because the pointwise maximum of convex functions lying below $h$ is a convex function, and the smallest concave overestimator of $h$, which is similarly well-defined). Then $Q = \{(x,y) \in [0,1]^n \times \mathbb{R} \mid -y = h(x)\}$ and $\mathrm{Conv}(Q) = \{(x,y) \in [0,1]^n \times \mathbb{R} \mid \underline{h}(x) \leqslant -y \leqslant \overline{h}(x)\}$ (e.g. [32, Proposition B.1]). Function $h$ is concave and hence $\overline{h} = h$, so it only remains to describe $\underline{h}$.

To describe $\underline{h}$, we define a set function $H$ based on $h$ (see Definition 2), which is submodular because $-\rho$ is concave and $w$ is non-negative (e.g. see [1, Section 3.1]). Submodularity allows us to describe the lower convex envelope of the continuous function $h$ through the *Lovász extension* of the set function $H$. This extension is the piecewise affine function from $[0,1]^n$ to $\mathbb{R}$ defined over the pieces $\{\mathbf{T}_\pi : \pi \in \mathcal{S}_n\}$, which equals $\max_{\pi \in \mathcal{S}_n}(-r_\pi)$ by convexity (e.g. see [6] for further details). Therefore the constraint required for $\mathrm{conv}(Q)$ is $\underline{h}(x) \leqslant -y \iff y \leqslant \min_{\pi \in \mathcal{S}_n} r_\pi(x)$ which completes the derivation of inequalities (14b) in the theorem statement. $\square$

Note that even though there are exponentially many inequalities in (14b), the tightest constraint on $y$ at any given point $x \in [0,1]^n$ can be efficiently found, by sorting the coordinates of $x$ to find the simplex $\mathbf{T}_\pi$ to which $x$ belongs. Moreover, going from $[0,1]^n$ to $[L, U]$ and eliminating the sign restriction on $w$ can be achieved with standard variable transformations (e.g. see the comments before [40, Corollary 3.14]).

Before demonstrating the variable transformations, we first further refine Theorem 2 for the case when $\rho$ is equal to the ReLU activation function $\sigma$. In particular, we generally have that each one of the $n!$ inequalities in (14b) is facet-defining because they hold at equality over the $n+1$ affinely independent points $\{(v, g(v))\}_{v \in \mathbf{V}_\pi}$. Hence, they are all needed to describe $\mathrm{Conv}(R)$. However, because it may happen that $r_\pi = r_{\pi'}$ for $\pi \neq \pi'$, the number of inequalities in (14b) after removing duplicates may be much smaller. The following lemma shows that this is indeed the case when $\rho$ is equal to the ReLU activation function $\sigma$. The lemma also gives a closed form expression for the interpolating functions $r_\pi$ in this case.

**Lemma 2.** *Let* $w \in \mathbb{R}_+^n$ *and* $-\sum_{i=1}^{n} w_i \leqslant b < 0$, $f(x) = w \cdot x + b$, *and* $g(x) = \sigma(f(x))$. *If* $\{ r_\pi \}_{\pi \in \mathcal{S}_n}$ *are the affine interpolation functions from Theorem 2, then*

$$\left\{ (x,y) \in \mathbb{R}^{n+1} \mid y \leqslant r_\pi(x) \quad \forall \pi \in \mathcal{S}_n \right\} = \left\{ (x,y) \in \mathbb{R}^{n+1} \mid y \leqslant r_{I,h}(x) \quad \forall (I,h) \in \mathcal{I} \right\}$$

*where* $\mathcal{I} \stackrel{\text{def}}{=} \left\{ (I,h) \in 2^{\llbracket n \rrbracket} \times \llbracket n \rrbracket \mid F(I) \geqslant 0, \ F(I \cup \{h\}) < 0 \right\}$, $r_{I,h}(x) \stackrel{\text{def}}{=} F(I)x_h + \sum_{i \in I} w_i x_i$, *and* $F : 2^{\llbracket n \rrbracket} \to \mathbb{R}$ *is the set-function associated to* $f$ *as defined in Definition 2.*

*Proof.* Fix $\pi \in \mathcal{S}_n$ and for each $j \in \llbracket n \rrbracket$ let $I(j) \stackrel{\text{def}}{=} \{ \pi(i) \}_{i=j+1}^{n}$.

Then the interpolation condition for $r_\pi$ given by $r_\pi(v) = g(v)$ for all $v \in \mathbf{V}_\pi$ is equivalent to

$$R_\pi(I(j)) = G(I(j)) \quad \forall j = 0, 1, \ldots, n \tag{15}$$

where $R_\pi$ and $G$ are the set-functions associated to $r_\pi$ and $g$ as defined in Definition 2. For $j = 0$, condition (15) implies $r_\pi(\mathbf{0}^n) = R_\pi(\llbracket n \rrbracket) = G(\llbracket n \rrbracket) = g(\mathbf{0}^n) = 0$ and hence there exists $\alpha \in \mathbb{R}^n$ such that $r_\pi(x) = \alpha \cdot x$ (i.e. $r_\pi$ is a linear function). For $j \in \llbracket n \rrbracket$, condition (15) further implies that

$$\alpha_{\pi(j)} = g\left( \sum_{i=0}^{j} e(\pi(i)) \right) - g\left( \sum_{i=0}^{j-1} e(\pi(i)) \right) = G(I(j)) - G(I(j-1)) \quad \forall j \in \llbracket n \rrbracket. \tag{16}$$

Now, because $F(\varnothing) = f(\mathbf{1}^n) \geqslant 0$, $w \in \mathbb{R}_+^n$, and $b < 0$, there exists a unique $k \in \llbracket n \rrbracket$ such that $(I(k), \pi(k)) \in \mathcal{I}$. Furthermore, $w \in \mathbb{R}_+^n$, $F(I(k) \cup \{ \pi(k) \}) = F(I(k-1)) < 0$, and $F(I(k)) \geqslant 0$ imply

$$F(I(j)) < 0 \quad \text{and} \quad G(I(j)) = 0 \qquad \forall j = 0, \ldots, k-1; \tag{17a}$$

$$F(I(j)) \geqslant 0 \quad \text{and} \quad G(I(j)) = F(I(j)) \quad \forall j = k, \ldots, n. \tag{17b}$$

Equations (16) and (17a) imply $\alpha_{\pi(j)} = 0$ for all $j \in \llbracket k \rrbracket$ or equivalently $\alpha_i = 0$ for all $i \notin I(k) \cup \{ \pi(k) \}$. Equations (16) and (17) imply $\alpha_{\pi(k)} = G(I(k)) = F(I(k))$. Finally, equations (16) and (17b) imply that $\alpha_{\pi(j)} = w_{\pi(j)}$ for all $j = k+1, \ldots, n$ or equivalently $\alpha_i = w_i$ for all $i \in I$. Hence, $r_\pi = r_{I(k), \pi(k)}$. The lemma follows by noting that for any $(I,h) \in \mathcal{I}$ there exists at least one $\pi \in \mathcal{S}_n$ such that $(I(k), \pi(k)) = (I,h)$. $\qquad\square$

Finally, we obtain the proof of Theorem 1 recalling that $f(x) = w \cdot x + b$ for $w \in \mathbb{R}^n$ and $b \in \mathbb{R}$, and $S = \{ (x,y) \in [L,U] \times \mathbb{R} \mid y = \sigma(f(x)) \}$ for $L, U \in \mathbb{R}^n$ such that $L < U$.

**Theorem 1.** *If* $\ell(\llbracket n \rrbracket) \geqslant 0$, *then* $\mathrm{Conv}(S) = S = \{ (x,y) \in [L,U] \times \mathbb{R} \mid y = f(x) \}$. *Alternatively, if* $\ell(\varnothing) < 0$, *then* $\mathrm{Conv}(S) = S = [L,U] \times \{ 0 \}$. *Otherwise,* $\mathrm{Conv}(S)$ *is equal to the set of all* $(x,y) \in \mathbb{R}^n \times \mathbb{R}$ *satisfying*

$$y \geqslant w \cdot x + b, \quad y \geqslant 0, \quad L \leqslant x \leqslant U \tag{6a}$$

$$y \leqslant \sum_{i \in I} w_i(x_i - \breve{L}_i) + \frac{\ell(I)}{\breve{U}_h - \breve{L}_h}(x_h - \breve{L}_h) \qquad \forall (I,h) \in \mathcal{J}. \tag{6b}$$

*Furthermore, if* $d \stackrel{\text{def}}{=} |\{ i \in \llbracket n \rrbracket \mid w_i \neq 0 \}|$, *then* $d \leqslant |\mathcal{J}| \leqslant \lceil \frac{1}{2}d \rceil \binom{d}{\lceil \frac{1}{2}d \rceil}$ *and for each of these inequalities (and each* $d \in \llbracket n \rrbracket$) *there exist data that makes it hold at equality.*

*Proof.* Recalling that $\mathcal{J} \stackrel{\text{def}}{=} \left\{ (I,h) \in 2^{\llbracket n \rrbracket} \times \llbracket n \rrbracket \mid \ell(I) \geqslant 0, \quad \ell(I \cup \{h\}) < 0, \quad w_i \neq 0 \quad \forall i \in I \right\}$ we can assume without loss of generality that $w_i \neq 0$ for all $i \in \llbracket n \rrbracket$ and hence $d = n$ (Indices $i$ with $w_i = 0$ do not affect (6b) or the definition of $\mathcal{J}$ and the only inequalities for $S$ or $\mathrm{Conv}(S)$ in which a given $x_i$ appears are $L_i \leqslant x_i \leqslant U_i$).

For the first case, the result follows because $f(x) < 0$ for all $x \in [L,U]$ and hence $g(x) = 0$ for all $x \in [L,U]$.

For the second case, the result follows because $f(x) \geqslant 0$ for all $x \in [L,U]$ and hence $g(x) = f(x)$ for all $x \in [L,U]$.

For the third case, recall that $\breve{L}_i = \begin{cases} L_i & w_i \geqslant 0 \\ U_i & \text{o.w.} \end{cases}$ and $\breve{U}_i = \begin{cases} U_i & w_i \geqslant 0 \\ L_i & \text{o.w.} \end{cases}$, and consider the affine variable transformation given by

$$\breve{x}_i \stackrel{\text{def}}{=} \frac{x_i - \breve{L}_i}{\breve{U}_i - \breve{L}_i} \quad \text{and} \quad x_i = (\breve{U}_i - \breve{L}_i)\breve{x}_i + \breve{L}_i \quad \forall i \in \llbracket n \rrbracket. \tag{18}$$

Let $\breve{w}_i \stackrel{\text{def}}{=} w_i(\breve{U}_i - \breve{L}_i)$ for each $i \in [\![n]\!]$, $\breve{b} \stackrel{\text{def}}{=} b + \sum_{i=1}^n w_i \breve{L}_i = \ell([\![n]\!]) < 0$, and $\breve{f}(\breve{x}) \stackrel{\text{def}}{=} \breve{w} \cdot \breve{x} + \breve{b}$ (recall that $\ell(I) \stackrel{\text{def}}{=} \sum_{i \in I} w_i \breve{L}_i + \sum_{i \notin I} w_i \breve{U}_i + b$). Then we may infer that

$$\breve{w}_i \breve{x}_i = w_i(x_i - \breve{L}_i) \quad \forall i \in [\![n]\!], \tag{19}$$

that $f(x) = \breve{f}(\breve{x})$, and finally that $(x, y) \in S$ if and only if $(\breve{x}, y) \in \breve{S} \stackrel{\text{def}}{=} \left\{ (\breve{x}, y) \in [0,1]^n \times \mathbb{R} \mid y = \sigma(\breve{f}(\breve{x})) \right\}$.

In addition, we conclude $\breve{w} \in \mathbb{R}_+^n$, using the definition of $\breve{L}$ and $\breve{U}$ and the fact that $L < U$. Hence, Theorem 2 and Lemma 2 are applicable for $\breve{S}$ and $\breve{g}(\breve{x}) = \sigma(\breve{f}(\breve{x}))$. Then

$$\text{Conv}(\breve{S}) = \left\{ (\breve{x}, y) \in [0,1]^n \times \mathbb{R}_+ \mid \breve{f}(\breve{x}) \leq y \leq r_{I,h}(\breve{x}) \quad \forall (I, h) \in \mathcal{I} \right\}$$

where $\mathcal{I} = \left\{ (I, h) \in 2^{[\![n]\!]} \times [\![n]\!] \mid \breve{F}(I) \geq 0, \; \breve{F}(I \cup \{h\}) < 0 \right\}$ and $r_{I,h}(\breve{x}) = \breve{F}(I)\breve{x}_h + \sum_{i \in I} \breve{w}_i \breve{x}_i$. Using the definitions of $\breve{b}$ and $\breve{w}_i$ we get

$$\breve{F}(I) = \sum_{i \notin I} \breve{w}_i + \breve{b} = \sum_{i \notin I} w_i \left( \breve{U}_i - \breve{L}_i \right) + \left( b + \sum_{i=1}^n w_i \breve{L}_i \right) = \sum_{i \notin I} w_i \breve{U}_i + \sum_{i \in I} w_i \breve{L}_i + b = \ell(I) \tag{20}$$

and hence $\mathcal{I} = \mathcal{J} = \left\{ (I, h) \in 2^{[\![n]\!]} \times [\![n]\!] \mid \ell(I) \geq 0, \; \ell(I \cup \{h\}) < 0 \right\}$. Combining (18–20), we get

$$\breve{r}_{I,h}(\breve{x}) = \breve{F}(I)\breve{x}_h + \sum_{i \in I} \breve{w}_i \breve{x}_i = \ell(I) \frac{x_h - \breve{L}_h}{\breve{U}_h - \breve{L}_h} + \sum_{i \in I} w_i \left( x_i - \breve{L}_i \right).$$

Hence, $\text{Conv}(S)$ is described by (6).

Finally, $(I, h) \in \mathcal{J}$ if and only if the hyperplane $\sum_{i=1}^n w_i x_i + b = 0$ cuts the edge $uv$ of $[0,1]^n$ given by $u \stackrel{\text{def}}{=} \sum_{i \notin (I \cup \{h\})} e(i)$ and $v \stackrel{\text{def}}{=} \sum_{i \notin I} e(i)$ (with the convention that an empty sum is equal to zero). The result on $|\mathcal{J}|$ then follows by Lemma 1 recalling that without loss of generality we have assumed $n = d$.

$\square$

## A.2 An alternative proof using mixed-integer programming and projection

We can alternatively prove Theorem 1 by connecting it to the MIP formulation from [3] for $S$ defined in (5). For this, first recall that that $f(x) = w \cdot x + b$ for $w \in \mathbb{R}^n$ and $b \in \mathbb{R}$, and $S = \{ (x, y) \in [L, U] \times \mathbb{R} \mid y = \sigma(f(x)) \}$ for $L, U \in \mathbb{R}^n$ such that $L < U$.

**Corollary 2.** *Let*

$$R_{\text{sharp}} \stackrel{\text{def}}{=} \left\{ (x, y, z) \in [L, U] \times \mathbb{R} \times [0,1]^2 \; \middle| \; \begin{array}{l} y \geq 0, \\ y \geq w \cdot x + b, \\ y \leq \bar{f}(x, z), \\ z_1 + z_2 = 1 \end{array} \right\},$$

*where*

$$\bar{f}(x, z) \stackrel{\text{def}}{=} \max_{\tilde{x}^1, \tilde{x}^2} \left\{ w \cdot \tilde{x}^2 + bz_2 \; \middle| \; \begin{array}{l} x = \tilde{x}^1 + \tilde{x}^2, \\ Lz_k \leq \tilde{x}^k \leq Uz_k \quad \forall k \in [\![2]\!] \\ \tilde{x}^1, \tilde{x}^2 \in \mathbb{R}^n \end{array} \right\}.$$

*Then* $\text{Conv}(S) = \text{Proj}_{x,y}(R_{\text{sharp}}) \stackrel{\text{def}}{=} \left\{ (x, y) \in \mathbb{R}^{n+1} \mid \exists z \in \mathbb{R}^2 \; s.t. \; (x, y, x) \in R_{\text{sharp}} \right\}.$

*Proof.* Follows from [3, Proposition 5] for the case $d = 2$, $w^1 = 0$, $b^1 = 0$, $w^2 = w$, $b^2 = b$. $\square$

**Lemma 3.** *Let*

$$R \overset{\text{def}}{=} \left\{ (x, y) \in [L, U] \times \mathbb{R} \ \middle| \ \begin{array}{l} y \geqslant 0, \\ y \geqslant w \cdot x + b, \\ y \leqslant \tilde{f}(x) \end{array} \right\}$$

*where*

$$\tilde{f}(x) \overset{\text{def}}{=} \max_{\tilde{x}^1, \tilde{x}^2, z} \left\{ w \cdot \tilde{x}^2 + bz_2 \ \middle| \ \begin{array}{c} x = \tilde{x}^1 + \tilde{x}^2, \\ Lz_k \leqslant \tilde{x}^k \leqslant Uz_k \qquad \forall k \in [\![2]\!] \\ \tilde{x}^1, \tilde{x}^2 \in \mathbb{R}^n \\ z_1 + z_2 = 1 \\ z \in [0, 1]^2 \end{array} \right\}. \tag{21}$$

*Then* $\mathrm{Conv}\,(S) = R$.

*Proof.* By Corollary 2 it suffices to show $R = \mathrm{Proj}_{x,y}(R_{\mathrm{sharp}})$.

Inclusion $\mathrm{Proj}_{x,y}(R_{\mathrm{sharp}}) \subseteq R$ follows by noting that $\bar{f}(\hat{x}, \hat{z}) \leqslant \tilde{f}(\hat{x})$ for any $(\hat{x}, \hat{y}, \hat{z}) \in R_{\mathrm{sharp}}$.

For inclusion $R \subseteq \mathrm{Proj}_{x,y}(R_{\mathrm{sharp}})$, let $(\hat{x}, \hat{y}) \in R$, and let $(\tilde{x}^1, \tilde{x}^2, z) \in \mathbb{R}^{2n+2}$ be an optimal solution to the optimization problem in the right hand side of (21) for $x = \hat{x}$. Such solution exists because for $\hat{x} \in [L, U]$ this optimization problem is the maximization of a linear function over a non-empty bounded polyhedron. Then, $\tilde{f}(\hat{x}) = \bar{f}(\hat{x}, z)$, and hence $(\hat{x}, \hat{y}, z) \in R_{\mathrm{sharp}}$. $\qquad \square$

**Theorem 1.** *If $\ell([\![n]\!]) \geqslant 0$, then $\mathrm{Conv}(S) = S = \{ (x, y) \in [L, U] \times \mathbb{R} \mid y = f(x) \}$. Alternatively, if $\ell(\varnothing) < 0$, then $\mathrm{Conv}(S) = S = [L, U] \times \{ 0 \}$. Otherwise, $\mathrm{Conv}(S)$ is equal to the set of all $(x, y) \in \mathbb{R}^n \times \mathbb{R}$ satisfying*

$$y \geqslant w \cdot x + b, \quad y \geqslant 0, \quad L \leqslant x \leqslant U \tag{6a}$$

$$y \leqslant \sum_{i \in I} w_i(x_i - \check{L}_i) + \frac{\ell(I)}{\check{U}_h - \check{L}_h}(x_h - \check{L}_h) \qquad \forall (I, h) \in \mathcal{J}. \tag{6b}$$

*Furthermore, if $d \overset{\text{def}}{=} |\{ i \in [\![n]\!] \mid w_i \neq 0 \}|$, then $d \leqslant |\mathcal{J}| \leqslant \lceil \frac{1}{2}d \rceil \binom{d}{\lceil \frac{1}{2}d \rceil}$ and for each of these inequalities (and each $d \in [\![n]\!]$) there exist data that makes it hold at equality.*

*Proof.* Recalling that $\mathcal{J} \overset{\text{def}}{=} \{ (I, h) \in 2^{[\![n]\!]} \times [\![n]\!] \mid \ell(I) \geqslant 0, \quad \ell(I \cup \{h\}) < 0, \quad w_i \neq 0 \quad \forall i \in I \}$ we can assume without loss of generality that $w_i \neq 0$ for all $i \in [\![n]\!]$ and hence $d = n$ (Indices $i$ with $w_i = 0$ do not affect (6b) or the definition of $\mathcal{J}$ and the only inequalities for $S$ or $\mathrm{Conv}(S)$ in which $x_i$ appear for such index are $L_i \leqslant x_i \leqslant U_i$).

For the first case, the result follows because $f(x) < 0$ for all $x \in [L, U]$ and hence $g(x) = 0$ for all $x \in [L, U]$.

For the second case, the result follows because $f(x) \geqslant 0$ for all $x \in [L, U]$ and hence $g(x) = f(x)$ for all $x \in [L, U]$.

For the third case, it suffices to show that

$$\tilde{f}(x) = \min_{(I,h) \in \mathcal{J}} \left\{ \sum_{i \in I} w_i(x_i - \check{L}_i) + \frac{\ell(I)}{\check{U}_h - \check{L}_h}(x_h - \check{L}_h) \right\}, \tag{22}$$

in which case, set $R$ from Lemma 3 is exactly the set described by (6). To show (22) we first simplify the optimization problem defining $\tilde{f}(x)$ by applying the simple substitutions $\tilde{x} \overset{\text{def}}{=} \tilde{x}^2 = x - \tilde{x}^1$ and $z \overset{\text{def}}{=} z_2 = 1 - z_1$:

$$\tilde{f}(x) = \max_{\tilde{x}, z} \left\{ w \cdot \tilde{x} + bz \ \middle| \ \begin{array}{c} L(1 - z) \leqslant x - \tilde{x} \leqslant U(1 - z), \\ Lz \leqslant \tilde{x} \leqslant Uz, \\ z \in [0, 1] \end{array} \right\}.$$

This optimization problem is feasible and bounded when $L \leqslant x \leqslant U$, and thus we may assume an optimal solution exists.

Consider some $i \in [\![n]\!]$. If $w_i > 0$, then $\tilde{x}_i \geqslant L_i z$ and $x_i - \tilde{x}_i \leqslant U_i(1-z)$ hold at any optimal solution, since we are maximizing the problem and each constraint involves only a single $x_i$ and $z$. Analogously, if $w_i < 0$, then $\tilde{x}_i \leqslant U_i z$ and $x_i - \tilde{x}_i \geqslant L_i(1-z)$ are implied as well. To unify these two cases into one as a simplification, observe that these constraints can be expressed as $w_i\tilde{x}_i \geqslant w_i\breve{L}_i z$ and $w_i(x_i - \tilde{x}_i) \leqslant w_i\breve{U}_i(1-z)$ respectively (recall that $w_i \neq 0$ by assumption, and that $\breve{L}_i = L_i$ if $w_i \geqslant 0$, or $U_i$ otherwise, and $\breve{U}_i = U_i$ if $w_i \geqslant 0$, or $L_i$ otherwise). Therefore, we can drop these constraints and keep the remaining ones:

$$\tilde{f}(x) = \max_{\tilde{x},z} \left\{ w \cdot \tilde{x} + bz \;\middle|\; \begin{array}{rl} w_i(x_i - \tilde{x}_i) \geqslant w_i\breve{L}_i(1-z) & \forall i \in [\![n]\!], \\ w_i\tilde{x}_i \leqslant w_i\breve{U}_i z & \forall i \in [\![n]\!] \\ z \in [0,1] \end{array} \right\}.$$

Define $\gamma_i \stackrel{\text{def}}{=} w_i(\breve{U}_i z - \tilde{x}_i)$ for all $i \in [\![n]\!]$. We can then rewrite the problem as:

$$\tilde{f}(x) = \max_{\gamma,z} \left\{ (w \cdot \breve{U} + b)z - \sum_{i=1}^{n} \gamma_i \;\middle|\; \begin{array}{rl} w_i(\breve{U}_i - \breve{L}_i)z - \gamma_i \leqslant w_i(x_i - \breve{L}_i) & \forall i \in [\![n]\!] \\ \gamma \geqslant 0, \\ z \in [0,1] \end{array} \right\}.$$

We next take the dual of this problem. By strong duality, the following holds:

$$\tilde{f}(x) = \min_{\alpha,\beta} \left\{ \sum_{i=1}^{n} w_i(x_i - \breve{L}_i)\alpha_i + \beta \;\middle|\; \begin{array}{l} \sum_{i=1}^{n} w_i(\breve{U}_i - \breve{L}_i)\alpha_i + \beta \geqslant \sum_{i=1}^{n} w_i\breve{U}_i + b, \\ \alpha \in [0,1]^n, \\ \beta \geqslant 0 \end{array} \right\}.$$

To conclude the proof, we describe the optimal solutions of the optimization problem above. Note that it is a minimization variant of a fractional knapsack problem and it can be solved by a greedy algorithm, in which we order the indices of $\alpha$ by $\frac{x_i - \breve{L}_i}{\breve{U}_i - \breve{L}_i}$ and maximally select those with the smallest ratios, until the knapsack constraint is satisfied at equality. We also need to consider $\beta$ in the knapsack, but since the ratios for $\alpha_i$ are in $[0,1]$ and the ratio for $\beta$ is $1$, $\beta$ would only be picked last. Moreover, under the assumptions of our current third case, we have $\ell([\![n]\!]) = \sum_{i=1}^{n} w_i\breve{L}_i + b < 0$, and thus that we can satisfy the knapsack constraint by choosing from $\alpha$'s (recall that $\ell(I) = \sum_{i \in I} w_i\breve{L}_i + \sum_{i \notin I} w_i\breve{U}_i + b$). Therefore we may set $\beta = 0$.

Let $I$ be the set of indices in which $\alpha_i = 1$ for the optimal solution and $h$ be the next index to be considered by the greedy procedure after the elements in $I$. Then

$$\alpha_h = \frac{\left(\sum_{i=1}^{n} w_i\breve{U}_i + b\right) - \left(\sum_{i \in I} w_i(\breve{U}_i - \breve{L}_i)\right)}{\breve{U}_h - \breve{L}_h} = \frac{\ell(I)}{\breve{U}_h - \breve{L}_h} \in [0,1).$$

Observe that $\ell(I) \geqslant 0$ is equivalent to stating that the items in $I$ are below the knapsack capacity, since $\ell(I)$ equals the capacity of the knapsack minus the total weight of the items in $I$. Therefore, $\ell(I) \geqslant 0$ and $\ell(I \cup \{h\}) < 0$ (i.e. the items in $I$ fit but we can only add $h$ partially). Hence, we can write the optimization problem defining $\tilde{f}(x)$ as finding the optimal $I$ and $h$:

$$\tilde{f}(x) = \min_{I, h \notin I} \left\{ \sum_{i \in I} w_i(x_i - \breve{L}_i) + \frac{\ell(I)}{\breve{U}_h - \breve{L}_h}(x_h - \breve{L}_h) \;\middle|\; \ell(I) \geqslant 0, \; \ell(I \cup \{h\}) < 0 \right\}.$$

We obtain (22) by recalling that $\mathcal{J} = \left\{ (I, h) \in 2^{[\![n]\!]} \times [\![n]\!] \mid \ell(I) \geqslant 0, \; \ell(I \cup \{h\}) < 0 \right\}$.

Finally, $(I, h) \in \mathcal{J}$ if and only if the hyperplane $\sum_{i=1}^{n} w_i x_i + b = 0$ cuts the edge $uv$ of $[0,1]^n$ given by $u \stackrel{\text{def}}{=} \sum_{i \notin (I \cup \{h\})} e(i)$ and $v \stackrel{\text{def}}{=} \sum_{i \notin I} e(i)$ (with the convention that an empty sum is equal to zero). The result on $|\mathcal{J}|$ then follows by Lemma 1 recalling that without loss of generality we have assumed $n = d$. $\qquad\square$

# B   Proofs of other results from Section 3

**Proposition 1.** *For any input dimension $n$, there exists a point $\tilde{x} \in \mathbb{R}^n$, and a problem instance given by the affine function $f$, the $\Delta$-relaxation $C_\Delta$, and the single neuron set $S$ such that $\left(\max_{y:(f(\tilde{x}),y)\in C_\Delta} y\right) - \left(\max_{y:(\tilde{x},y)\in \mathrm{Conv}(S)} y\right) = \Omega(n)$.*

*Proof.* This follows as a straightforward extension of [3, Example 2], as the $\Delta$-relaxation is equal to the projection of the big-$M$ formulation presented in that work. □

The following proposition shows how the additional structure in Lemma 2 allows increasing the speed of checking for violated inequalities from $\mathcal{O}(n\log(n))$, achievable by sorting the input components, to $\mathcal{O}(n)$.

**Proposition 2.** *The optimization problem (7) can be solved in $\mathcal{O}(n)$ time.*

*Proof.* Recall that $\mathcal{J} \overset{\text{def}}{=} \left\{ (I,h) \in 2^{[\![n]\!]} \times [\![n]\!] \mid \ell(I) \geqslant 0, \quad \ell(I \cup \{h\}) < 0, \quad w_i \neq 0 \; \forall i \in I \right\}$, $\ell(I) \overset{\text{def}}{=} \sum_{i\in I} w_i \breve{L}_i + \sum_{i\notin I} w_i \breve{U}_i + b$, $\breve{L}_i \overset{\text{def}}{=} \begin{cases} L_i & w_i \geqslant 0 \\ U_i & \text{o.w.} \end{cases}$ and $\breve{U}_i \overset{\text{def}}{=} \begin{cases} U_i & w_i \geqslant 0 \\ L_i & \text{o.w.} \end{cases}$ for each $i \in [\![n]\!]$, and (7) is the optimization problem given by

$$v(x) \overset{\text{def}}{=} \min \left\{ \sum_{i\in I} w_i(x_i - \breve{L}_i) + \frac{\ell(I)}{\breve{U}_h - \breve{L}_h}(x_h - \breve{L}_h) \;\middle|\; (I,h) \in \mathcal{J} \right\}.$$

First, we can check in $\mathcal{O}(n)$ time if $\ell([\![n]\!]) \geqslant 0$ or $\ell(\varnothing) < 0$, in which case $\mathcal{J} = \varnothing$ and (7) is infeasible. Otherwise, $\ell([\![n]\!]) < 0$, $\ell(\varnothing) \geqslant 0$, and $\mathcal{J} \neq \varnothing$.

We can also remove in $\mathcal{O}(n)$ time all $i \in [\![n]\!]$ such that $w_i = 0$. Then without loss of generality we may assume that $w_i \neq 0$ for all $i \in [\![n]\!]$ and hence $L < U$ implies that

$$w_i(\breve{U}_i - \breve{L}_i) > 0 \quad \forall i \in [\![n]\!]. \tag{23}$$

We will show that (7) is equivalent to the linear programming problem

$$\omega(x) \overset{\text{def}}{=} \min_v \quad \sum_{i=1}^{n} w_i(x_i - \breve{L}_i)v_i \tag{24a}$$

$$\text{s.t.} \quad \sum_{i=1}^{n} w_i(\breve{U}_i - \breve{L}_i)v_i = \sum_{i=1}^{n} w_i\breve{U}_i + b, \tag{24b}$$

$$0 \leqslant v \leqslant 1. \tag{24c}$$

Note that the set of basic feasible solutions for the linear programming problem is exactly the set of all feasible points with at most one fractional component (see, e.g., [7, Chapter 3]). That is, all basic feasible solutions of (24) are elements of $\mathcal{V} \overset{\text{def}}{=} \left\{ v \in [0,1]^n \mid |\{ i \in [\![n]\!] \mid v_i \in (0,1) \}| \leqslant 1 \right\}$.

To prove that $\omega(x) \leqslant v(x)$, consider the mapping $\Phi : \mathcal{J} \to \mathcal{V}$ given by

$$\Phi\left((I,h)\right)_i = \begin{cases} 1 & i \in I \\ \frac{\ell(I)}{w_h(\breve{U}_h - \breve{L}_h)} & i = h \quad \forall i \in [\![n]\!]. \\ 0 & \text{o.w.} \end{cases}$$

Let $(\bar{I}, \bar{h}) \in \mathcal{J}$ be an optimal solution for (7) and let $\bar{v} = \Phi\left((\bar{I},\bar{h})\right)$. Then

$$\sum_{i=1}^{n} w_i(\breve{U}_i - \breve{L}_i)\bar{v}_i = \sum_{i\in\bar{I}} w_i(\breve{U}_i - \breve{L}_i) + w_{\bar{h}}(\breve{U}_{\bar{h}} - \breve{L}_{\bar{h}})\frac{\ell(\bar{I})}{w_{\bar{h}}(\breve{U}_{\bar{h}} - \breve{L}_{\bar{h}})} = \sum_{i=1}^{n} w_i\breve{U}_i + b,$$

and hence $\bar{v}$ satisfies (24b). Algebraic manipulation shows that

$$w_h(\breve{U}_h - \breve{L}_h) = \ell(I) - \ell(I \cup \{h\}) \quad \forall I \subseteq [\![n]\!], \quad h \in [\![n]\!] \setminus I. \tag{25}$$

In addition, $(\bar{I}, \bar{h}) \in \mathcal{J}$ implies $\ell(\bar{I}) \geqslant 0$ and $\ell(\bar{I} \cup \{\bar{h}\}) < 0$. Combining this with (25) gives the inequality $\ell(\bar{I}) < w_{\bar{h}}(\breve{U}_{\bar{h}} - \breve{L}_{\bar{h}})$. Therefore, $\bar{v}_h \in [0, 1)$, and hence $\bar{v}_h$ is feasible for (24). In addition, for any $(I, h) \in \mathcal{J}$ we have that

$$\sum_{i=1}^{n} w_i(x_i - \breve{L}_i)\bar{v}_i = \sum_{i \in I} w_i(x_i - \breve{L}_i) + w_h(\breve{U}_h - \breve{L}_h)\frac{\ell(I)}{w_h(x_h - \breve{L}_h)}$$

and hence the objective value of $\bar{v}_h$ for (24) is the same as the objective value of $(I, h)$ for (7).

To prove $\omega(x) \geqslant \upsilon(x)$ we will show that, through $\Phi$, the greedy procedure to solve (7) described in the main text just before the statement of Proposition 2, becomes the standard greedy procedure for (24) and hence also yields an optimal basic feasible solution to (24). For simplicity, assume without loss of generality that we have re-ordered the indices in $[\![n]\!]$ so that

$$\frac{x_1 - \breve{L}_1}{\breve{U}_1 - \breve{L}_1} \leqslant \frac{x_2 - \breve{L}_2}{\breve{U}_2 - \breve{L}_2} \leqslant \cdots \leqslant \frac{x_n - \breve{L}_n}{\breve{U}_n - \breve{L}_n}. \tag{26}$$

Then the greedy procedure that incrementally grows $I$ terminates with some $(I, h) \in \mathcal{J}$ where $I = [\![h-1]\!]$. Then $v = \Phi(([\![h-1]\!], h))$ is a basic feasible solution for (24) with the same objective value as the objective value of $(I, h)$ for (7). To conclude that $\omega(x) \geqslant \upsilon(x)$, we claim that $v$ is an optimal solution for (24) since the standard greedy procedure for (24) is known to generate the optimal solution for this problem. For completeness, we give the following self contained proof of the claim. Assume for a contradiction that $\omega(x) < \sum_{i=1}^{n} w_i(x_i - \breve{L}_i)v_i$ and let $v'$ be an optimal solution to (24). Because $v' \neq v$ and both $v$ and $v'$ satisfy (24b), (23) implies there must exists $j_1, j_2 \in [\![n]\!]$ such that $j_1 < j_2, j_1 \leqslant h, v'_{j_1} < v_{j_1}, j_2 \geqslant h$ and $v'_{j_2} > v_{j_2}$. Let $\epsilon > 0$ be the largest value such that $v'_{j_1} + \frac{\epsilon}{w_{j_1}(\breve{U}_{j_1} - \breve{L}_{j_1})} \leqslant v_{j_1}$ and $v'_{j_2} - \frac{\epsilon}{w_{j_2}(\breve{U}_{j_2} - \breve{L}_{j_2})} \geqslant v_{j_2}$, and let

$$v'' \overset{\text{def}}{=} v' + \frac{\epsilon}{w_{j_1}(\breve{U}_{j_1} - \breve{L}_{j_1})}e(j_1) - \frac{\epsilon}{w_{j_2}(\breve{U}_{j_2} - \breve{L}_{j_2})}e(j_2).$$

By (26) we either have

$$\frac{x_{j_1} - \breve{L}_{j_1}}{\breve{U}_{j_1} - \breve{L}_{j_1}} = \frac{x_{j_2} - \breve{L}_{j_2}}{\breve{U}_{j_2} - \breve{L}_{j_2}} \quad \text{or} \quad \frac{x_{j_1} - \breve{L}_{j_1}}{\breve{U}_{j_1} - \breve{L}_{j_1}} < \frac{x_{j_2} - \breve{L}_{j_2}}{\breve{U}_{j_2} - \breve{L}_{j_2}}. \tag{27}$$

In the first case $v''$ is a feasible solution to (24) that has fewer different components with $v$ and has the same objective value as $v'$. Hence, by repeating this procedure we will eventually have the second case in which $v''$ is a feasible solution to (24) that has an objective value strictly smaller than that of $v'$, which contradicts the optimality of $v'$.

The greedy procedure to solve (7) and (24) can be executed in $\mathcal{O}(n \log(n))$ time through the sorting required to get (26). However, an optimal basic feasible solution $\hat{\alpha}$ to (24) can also be obtained in $\mathcal{O}(n)$ time by solving a weighted median problem (e.g. [21, Chapter 17.1]). This solution can be converted to an optimal solution to (7) in $\mathcal{O}(n)$ time as follows. Because $\hat{\alpha}$ is a basic feasible solution to (24), it has at most one fractional component (see, e.g., [7, Chapter 3]). Take $\hat{I} = \{ i \in [\![n]\!] \mid \hat{\alpha}_i = 1 \}$. If $\hat{v}$ has one fractional component, take $\hat{h}$ to be this component. Then, because $\hat{\alpha}$ satisfies (24b) we have

$$w_{\hat{h}}(\breve{U}_{\hat{h}} - \breve{L}_{\hat{h}})\hat{\alpha}_{\hat{h}} = \sum_{i=1}^{n} w_i\breve{U}_i + b - \sum_{i \in \hat{I}} w_i(\breve{U}_i - \breve{L}_i) = \ell(\hat{I}) \tag{28}$$

Together with $\hat{\alpha}_{\hat{h}} \in (0, 1)$, (25) for $I = \hat{I}$ and $h = \hat{h}$, and (23) for $i = \hat{h}$, we have $\ell(\hat{I}) > 0$ and

$$\ell(\hat{I}) - \ell(\hat{I} \cup \{\hat{h}\}) > \ell(\hat{I}).$$

Then $\ell(\hat{I} \cup \{\hat{h}\}) < 0$ and $(\hat{I}, \hat{h}) \in \mathcal{J}$. Finally, (28) implies that the objective value of $\hat{\alpha}$ for (24) is the same as the objective value of $(\hat{I}, \hat{h})$ for (7).

If, on the other hand, $\hat{v}$ has no fractional component, then $\hat{\alpha}$ satisfying (24b) implies

$$0 = \sum_{i=1}^{n} w_i\breve{U}_i + b - \sum_{i \in \hat{I}} w_i(\breve{U}_i - \breve{L}_i) = \ell(\hat{I}). \tag{29}$$

Then, $\ell(\llbracket n \rrbracket) < 0$ implies that there exists $\hat{h} \in \llbracket n \rrbracket \setminus \hat{I}$ such that $\ell(\hat{I} \cup \{\hat{h}\}) < 0$ and $(\hat{I}, \hat{h}) \in \mathcal{J}$. Finally, (29) implies that the objective value of $\hat{\alpha}$ for (24) is the same as the objective value of $(\hat{I}, \hat{h})$ for (7). This conversion of an optimal basic feasible solution for (24) to a solution to (7) also gives an alternate proof to $\omega(x) \geqslant \upsilon(x)$. $\qquad\square$

**Corollary 1.** *If the weights $w$ and biases $b$ describing the neural network are rational, then the single-neuron relaxation* (4) *can be solved in polynomial time on the encoding sizes of $w$ and $b$.*

*Proof.* If $w$ and $b$ are rational, then the coefficients of the inequalities in (6b) are also rational numbers with sizes that are polynomial in the sizes of $w$ and $b$. Then the result follows from Proposition 2 and [13, Theorem 7.26]. $\qquad\square$

## C  Propagation algorithms

### C.1  Description and analysis of algorithms

In this section, we provide pseudocode for the propagation-based algorithms described in Section 4. In the scope of a single neuron, Algorithm 1 specifies the framework outlined in Section 4.1 and Algorithm 3 (which requires Algorithm 2) details our new algorithm proposed in Section 4.3. Finally, Algorithm 4 establishes how to compute bounds for the entire network, considering `DeepPoly` [36] and `Fast-Lin` [44] as possible initial methods.

---

**Algorithm 1** The Backwards Pass for Upper Bounds

---

1: **Inputs:**
    Input domain $X \subseteq \mathbb{R}^m$, affine functions $\mathcal{L}_i(z_{1:i-1}) = \sum_{j=1}^{i-1} w_{ij}^l z_j + b_i^l$,
    $\mathcal{U}_i(z_{1:i-1}) = \sum_{j=1}^{i-1} w_{ij}^u z_j + b_i^u$ for each $i = m+1, \ldots, \eta$, and affine function
    $\mathcal{C}(z) = \sum_{i=1}^{\eta} c_i z_i + b$
2: **Outputs:**
    Upper bound on $\mathcal{C}(z)$, optimal point $x^* \in X$, and boolean vector
    $(\texttt{ub\_used}_{m+1}, \ldots, \texttt{ub\_used}_\eta)$
3: **function** PROPAGATIONBOUND($X, \mathcal{L}, \mathcal{U}, \mathcal{C}$)
4:     $\texttt{ub\_used}_i \leftarrow \texttt{false}$ for all $i = m+1, \ldots, \eta$
5:     $Q \leftarrow \{\, i \mid c_i \neq 0, i > m \,\}$            $\triangleright$ Set of variable indices to be substituted
6:     $\texttt{expr} \leftarrow \sum_{i=1}^{\eta} c_i z_i + b$       $\triangleright$ Denote by $\texttt{expr.w[i]}$ the coefficient for $z_i$ in $\texttt{expr}, \forall i$
7:     **while** $Q$ is not empty **do**
8:         $i \leftarrow$ pop largest value from $Q$, removing it
9:         $\texttt{ub\_used}_i \leftarrow (\texttt{expr.w[i]} > 0)$
10:        $\texttt{expr} \leftarrow \texttt{expr} - \texttt{expr.w[i]}\, z_i$            $\triangleright$ Remove term from expression
11:        **if** $\texttt{expr.w[i]} > 0$ **then**
12:           $\texttt{expr} \leftarrow \texttt{expr} + \texttt{expr.w[i]}\, \mathcal{U}_i(z_{1:i-1})$
13:           $Q \leftarrow Q \cup \{\, j \mid w_{ij}^u \neq 0, j > m \,\}$
14:        **else if** $\texttt{expr.w[i]} < 0$ **then**
15:           $\texttt{expr} \leftarrow \texttt{expr} + \texttt{expr.w[i]}\, \mathcal{L}_i(z_{1:i-1})$
16:           $Q \leftarrow Q \cup \{\, j \mid w_{ij}^l \neq 0, j > m \,\}$
17:        **end if**
18:     **end while**
19:     $B, x^* \leftarrow \max_{x \in X} \texttt{expr}$, along with an optimal solution
20:     **return** $B, x^*, \texttt{ub\_used}$
21: **end function**

---

---

**Algorithm 2** The Forward Pass

---

1: **Inputs:**
    Partial optimal solution $x^* \in X$ from Algorithm 1 and boolean vector
    $(\texttt{ub\_used}_{m+1}, \ldots, \texttt{ub\_used}_\eta)$ where $\texttt{ub\_used}_i = \texttt{true}$ if the upper bound $\mathcal{U}_i$
    was used to substitute variable $i$ in Algorithm 1, or $\texttt{false}$ otherwise.
2: **Outputs:**
    Optimal solution $z^*_{1:\eta}$ to (9)

3: **function** RECOVERCOMPLETESOLUTION($x^*$, $\texttt{ub\_used}$)
4:     $z^*_i = x^*_i$ for $i = 1, \ldots, m$
5:     **for** $i = m + 1, \ldots, \eta$ **do**
6:         **if** $\texttt{ub\_used}_i = \texttt{true}$ **then**
7:             $z^*_i \leftarrow \mathcal{U}_i(z^*_{1:i-1})$
8:         **else**
9:             $z^*_i \leftarrow \mathcal{L}_i(z^*_{1:i-1})$
10:         **end if**
11:     **end for**
12:     **return** $z^*_{1:\eta}$
13: **end function**

---

---

**Algorithm 3** The Iterative Algorithm

---

1: **Inputs:**
    Input domain $X \subseteq \mathbb{R}^m$, initial affine bounding functions $\{\mathcal{L}_i^{\text{init}}\}_{i=m+1}^{\eta}$,
    $\{\mathcal{U}_i^{\text{init}}\}_{i=m+1}^{\eta}$, affine function $\mathcal{C} : \mathbb{R}^\eta \to \mathbb{R}$, and number of iterations $T \geqslant 0$
2: **Outputs:**
    An upper bound on $\max_{x \in X} \mathcal{C}(x)$

3: **function** TIGHTENEDPROPAGATIONBOUND($X, \mathcal{L}, \mathcal{U}, \mathcal{C}, k$)
4:     $\{\mathcal{L}_i, \mathcal{U}_i\}_{i=m+1}^{\eta} \leftarrow \{\mathcal{L}_i^{\text{init}}, \mathcal{U}_i^{\text{init}}\}_{i=m+1}^{\eta}$
5:     $B, x^*, \texttt{ub\_used} \leftarrow$ PROPAGATIONBOUND($X, \{\mathcal{L}_i\}_{i=m+1}^{\eta}, \{\mathcal{U}_i\}_{i=m+1}^{\eta}, \mathcal{C}$)
6:     **for** $\texttt{iter} = 1, \ldots, T$ **do**
7:         $z^*_{1:\eta} \leftarrow$ RECOVERCOMPLETESOLUTION($x^*$, $\texttt{ub\_used}$)
8:         **for** $i = m + 1, \ldots, \eta$ **do**
9:             $\mathcal{U}_i', v \leftarrow$ most violated inequality w.r.t. $z^*_{1:\eta}$ from (6b) (per Prop. 2) and its violation
10:             **if** $v > 0$ **then** $\mathcal{U}_i \leftarrow \mathcal{U}_i'$ **end if**
11:         **end for**
12:         $B', x^*, \texttt{ub\_used} \leftarrow$ PROPAGATIONBOUND($X, \{\mathcal{L}_i\}_{i=m+1}^{\eta}, \{\mathcal{U}_i\}_{i=m+1}^{\eta}, \mathcal{C}$)
13:         **if** $B' < B$ **then** $B \leftarrow B'$ **end if**
14:     **end for**
15:     **return** $B$
16: **end function**

---

**Proposition 4.** *The solution $z^*$ returned by Algorithm 2 is optimal for the relaxed problem* (9).

*Proof.* Denote by $\texttt{expr}^k \overset{\text{def}}{=} \sum_{j \in J^k} \mathtt{w}_j^k z_j + \mathtt{b}^k$ the expression $\texttt{expr}$ at the end of iteration $k = 1, \ldots, K$ of the while loop in Algorithm 1, for some subsets $J^1, \ldots, J^K \subseteq [\![\eta]\!]$, and let $\texttt{expr}^0$ be the initial $\texttt{expr}$ as defined in line 5, i.e. $\mathcal{C}(z)$. For each $k = 0, \ldots, K - 1$, we obtain $\texttt{expr}^{k+1}$ by replacing, for some $i$, $z_i$ by $\mathcal{U}_i(z_{1:i-1})$ if $\mathtt{w}_i^k > 0$, or by $\mathcal{L}_i(z_{1:i-1})$ if if $\mathtt{w}_i^k < 0$. Note that if $\mathtt{w}_i^k = 0$, we can safely ignore any substitution because it will not affect the expression. Due to the constraints (9c), this substitution implies that $\texttt{expr}^k \leqslant \texttt{expr}^{k+1}$ for any $z_{1:m} \in X$. This inductively establishes that, restricting to $z_{1:m} \in X$,

$$\mathcal{C}(z) = \sum_{j=1}^{\eta} c_j z_j + b \leqslant \sum_{j \in J^1} \mathtt{w}_j^1 z_j + \mathtt{b}^1 \leqslant \ldots \leqslant \sum_{j \in J^K} \mathtt{w}_j^K z_j + \mathtt{b}^K, \tag{30}$$

Note that $J^K \subseteq \{z_1, \ldots, z_m\}$ since we have made all the substitutions possible for $i > m$. Therefore, the optimal value of (9) is upper-bounded by the bound corresponding to the solution returned by

---

**Algorithm 4** FastC2V Algorithm

---

1: **Inputs:**
   A feedforward neural network as defined in (1) (with input domain $X$, ReLU neurons $i = m + 1, \ldots, N$, and a single affine output neuron indexed by $N + 1$), `initial_method` $\in \{$`DeepPoly`, `Fast-Lin`$\}$, and number of iterations per neuron $T \geqslant 0$ (note that if $T = 0$, we recover `DeepPoly` or `Fast-Lin`)

2: **Outputs:**
   Lower and upper bounds $\{\hat{L}_i, \hat{U}_i\}_{i=1}^{N+1}$ on the pre-activation function (if ReLU) or output (if affine) of neuron $i$

3: **function** FASTC2V($X, W, b,$ `initial_method`, $T$)
4:    **for** $i = m + 1, \ldots, N + 1$ **do**
5:       $\mathcal{C}(z_{1:i-1}) \leftarrow \sum_{j=1}^{i-1} w_{i,j} z_j + b_i$
6:       $\hat{L}_i \leftarrow -$ TIGHTENEDPROPAGATIONBOUND($X, \{\mathcal{L}_j\}_{j=m+1}^{i-1}, \{\mathcal{U}_j\}_{j=m+1}^{i-1}, -\mathcal{C}, T$)
7:       $\hat{U}_i \leftarrow$ TIGHTENEDPROPAGATIONBOUND($X, \{\mathcal{L}_j\}_{j=m+1}^{i-1}, \{\mathcal{U}_j\}_{j=m+1}^{i-1}, \mathcal{C}, T$)
8:       **if** $i = N + 1$ **then break end if**
9:       $\triangleright$ Build bounding functions $\mathcal{L}_i$ and $\mathcal{U}_i$ for subsequent iterations
10:       **if** $\hat{L}_i \geqslant 0$ **then**                $\triangleright$ ReLU $i$ is always active for any $z_{1:m} \in X$
11:          $\mathcal{L}_i(z_{1:i-1}) \leftarrow \sum_{j=1}^{i-1} w_{i,j} z_j + b_i$
12:          $\mathcal{U}_i(z_{1:i-1}) \leftarrow \sum_{j=1}^{i-1} w_{i,j} z_j + b_i$
13:       **else if** $\hat{U}_i \leqslant 0$ **then**          $\triangleright$ ReLU $i$ is always inactive for any $z_{1:m} \in X$
14:          $\mathcal{L}_i(z_{1:i-1}) \leftarrow 0$
15:          $\mathcal{U}_i(z_{1:i-1}) \leftarrow 0$
16:       **else**
17:          $\mathcal{U}_i(z_{1:i-1}) \leftarrow \frac{\hat{U}_i}{\hat{U}_i - \hat{L}_i}(\sum_{j=1}^{i-1} w_{i,j} z_j + b_i - \hat{L}_i)$
18:          **if** `initial_method` $=$ `DeepPoly` **then**
19:             **if** $|\hat{L}_i| \geqslant |\hat{U}_i|$ **then** $\mathcal{L}_i(z_{1:i-1}) \leftarrow 0$ **else** $\mathcal{L}_i(z_{1:i-1}) \leftarrow \sum_{j=1}^{i-1} w_{i,j} z_j + b_i$ **end if**
20:          **else**   $\triangleright$ `initial_method` $=$ `Fast-Lin`
21:             $\mathcal{L}_i(z_{1:i-1}) \leftarrow \frac{\hat{U}_i}{\hat{U}_i - \hat{L}_i}(\sum_{j=1}^{i-1} w_{i,j} z_j + b_i)$
22:          **end if**
23:       **end if**
24:    **end for**
25:    **return** $\{\hat{L}_i, \hat{U}_i\}_{i=1}^{N+1}$
26: **end function**

---

Algorithm 1, that is,

$$\max \left\{ \mathcal{C}(z) \mid z_{1:m} \in X, \text{ (9c) } \right\} \leqslant \max \left\{ \sum_{j \in J^K} \mathtt{w}_j^K z_j + \mathtt{b}^K \,\middle|\, z_{1:m} \in X \right\}.$$

To see that this upper bound is achieved, observe that each inequality in (30) holds as equality if we substitute $z_j = z_j^*$ for all $j$, by construction of Algorithm 2 and boolean vector ($\mathtt{ub\_used}_{m+1}, \ldots, \mathtt{ub\_used}_\eta$). Note also that $z^*$ satisfies (9c) by construction. That is, we have a feasible $z^*$ such that $\mathcal{C}(z^*)$ is no less than the optimal value of (9), and thus $z^*$ must be an optimal solution. $\qquad\square$

We would like to highlight to the interested reader that this result can also be derived from an argument using Fourier-Motzkin elimination [7, Chapter 2.8] to project out the intermediate variables $z_{m+1:\eta}$. Notably, as each inequality neuron has exactly one inequality upper bounding and one inequality lower bounding its post-activation value, this projection does not produce an "explosion" of new inequalities as is typically observed when applying Fourier-Motzkin to an arbitrary polyhedron.

Define $C \stackrel{\text{def}}{=} |\{ i \in [\![\eta]\!] \mid c_i \neq 0 \}|$ and suppose that we use the affine bounding inequalities from `Fast-Lin` or `DeepPoly`. Let $T$ be the number of iterations in Algorithm 3, $opt(X)$ be the time required to maximize an arbitrary affine function over $X$, and $A$ be the number of arcs in the network (i.e. nonzero weights).

(a) Structure and weights of the example network.

(b) Output of the example network (rotated).

Figure 4: An example network with 4 ReLUs on which we simulate `FastC2V`.

**Observation 1.** *Algorithm 1 runs in* $opt(X) + \mathcal{O}(C + A)$ *time. Algorithm 2 runs in* $\mathcal{O}(A)$ *time. Algorithm 3 runs in* $(T + 1)opt(X) + \mathcal{O}(T(C + A))$ *time.*

**Observation 2.** *Algorithm 4 takes* $\mathcal{O}(NT(opt(X) + A))$ *time if* $T \geqslant 1$. *If* $T = 0$, *then Algorithm 4 takes* $\mathcal{O}(N(opt(X) + A))$ *time.*

### C.2 Proofs of other results from section 4

**Proposition 3.** *The optimal value of* (9) *is no less than* $\max_{x \in X} \mathcal{C}\left(z_{1:\eta}\left(x\right)\right)$.

*Proof.* For any $x \in X$, by definition of validity in (8), setting $z_i \leftarrow z_i(x)$ for all $i = 1, \ldots, N$ yields a feasible solution to (9) with objective value $c\left(z_{1:N}\left(x\right)\right)$, completing the proof. □

## D An example for `FastC2V`

In this section, we walk through the `FastC2V` algorithm step-by-step for the following $\mathbb{R}^2 \to \mathbb{R}$ network with four ReLUs, also illustrated in Figure 4:

$$
\begin{aligned}
x_1 &\in [-1, 1] \\
x_2 &\in [-1, 1] \\
h_{11} &= \max(0, -x_1 + x_2 + 1) \\
h_{12} &= \max(0, -x_1 + 0.5) \\
h_{21} &= \max(0, h_{12} + 1) \\
h_{22} &= \max(0, -1.5h_{11} + h_{12} + 0.5) \\
y &= h_{21} + h_{22}
\end{aligned}
$$

Our goal is to compute an upper bound for $y$ using `FastC2V`.

Our procedure requires lower and upper bounds for each pre-activation function, and this can be obtained by running the same algorithm for each neuron, layer by layer. For simplicity, in this example we start from bounds computed via interval arithmetic.

Denote by $\hat{h}_{ij}$ the pre-activation function of $h_{ij}$. To apply interval arithmetic, we simply substitute the variables by their lower or upper bounds as to minimize or maximize them (applying the ReLU activation function when needed). For example, the interval arithmetic upper bound of $\hat{h}_{11}$ is $-1.5 \times (-1) + 1 + 0.5 = 3$. Starting at $x_1 \in [-1, 1]$ and $x_2 \in [-1, 1]$, we have:

$$
\begin{array}{llll}
\hat{h}_{11} \in [-1, 3] & (h_{11} \in [0, 3]) & \hat{h}_{21} \in [1, 2.5] & (h_{21} \in [1, 2.5]) \\
\hat{h}_{12} \in [-0.5, 1.5] & (h_{12} \in [0, 1.5]) & \hat{h}_{22} \in [-4, 2] & (h_{22} \in [0, 2])
\end{array}
$$

Note that $\hat{h}_{21} \geqslant 0$ for any input $x \in [-1, 1]^2$, and thus we may infer that the ReLU will always be active. That is, we can assume $h_{21} = h_{12} + 1$. This linearization step not only can have a large impact in bound strength, but also is required for correctness, as the formulations assume that the lower bound is negative and the upper bound positive.

Therefore, we drop $h_{21}$ altogether and set

$$y = h_{12} + h_{22} + 1.$$

Observe that interval arithmetic already gives us a simple upper bound on $y$ of 4.5.

We begin by applying `DeepPoly` [36] (or `CROWN-Ada` [49]), following Algorithm 1. Consider briefly a ReLU $y = \max(0, w^\top x + b)$ with pre-activation bounds $[\hat{L}, \hat{U}]$. In `DeepPoly`, we select the lower bounding inequality to be $y \geqslant 0$ if $|\hat{L}| \geqslant |\hat{U}|$, or $y \geqslant w^\top x + b$ otherwise. The upper bounding inequality comes from the $\Delta$-relaxation and can be expressed as $y \leqslant \frac{\hat{U}}{\hat{U}-\hat{L}}(w^\top x + b - \hat{L})$. Thus, based on the previously computed bounds, we have:

$$-x_1 + x_2 + 1 \leqslant h_{11} \leqslant -\frac{3}{4}x_1 + \frac{3}{4}x_2 + \frac{3}{2}$$

$$-x_1 + \frac{1}{2} \leqslant h_{12} \leqslant -\frac{3}{4}x_1 + \frac{3}{4}$$

$$0 \leqslant h_{22} \leqslant -\frac{1}{2}h_{11} + \frac{1}{3}h_{12} + \frac{3}{2}$$

The next step is to maximize $y$ over the relaxation given by the above inequalities plus bounds (including on the input). We replace variables with the above bounding inequalities, layer by layer. Since we are maximizing, we use the upper bounding inequality if the corresponding coefficient is positive, or the lower bound inequality otherwise. This maintains validity of the inequality throughout the process.

$$y = h_{12} + h_{22} + 1 \leqslant h_{12} + \left(-\frac{1}{2}h_{11} + \frac{1}{3}h_{12} + \frac{3}{2}\right) + 1$$

$$= -\frac{1}{2}h_{11} + \frac{4}{3}h_{12} + \frac{5}{2}$$

$$\leqslant -\frac{1}{2}(-x_1 + x_2 + 1) + \frac{4}{3}\left(-\frac{3}{4}x_1 + \frac{3}{4}\right) + \frac{5}{2}$$

$$= -\frac{1}{2}x_1 - \frac{1}{2}x_2 + 3$$

Now that we have inferred the above upper bounding inequality on $y$, we convert it into an upper bound by solving the simple problem $\max_{x \in [-1,1]^2} -\frac{1}{2}x_1 - \frac{1}{2}x_2 + 3$, which yields 4, with an optimal solution $(-1, -1)$. This is the resulting upper bound from the `DeepPoly` algorithm.

We next show how to tighten it with `FastC2V`. The first step is to recover an actual optimal solution of the relaxation above. This is the forward pass described in Algorithm 2.

We first make note that we used the upper bounding inequality for $h_{12}$ and $h_{22}$ and the lower bounding inequality for $h_{11}$. We start from the optimal solution in the input space, $(-1, -1)$, and recover values for each $h_{ij}$ and $y$ according to the bounding inequalities used, considering them to be equalities. For example, $\bar{h}_{11} = -(-1) + (-1) + 1 = 1$. The result is the solution $\bar{p} = (-1, -1, 1, 1.5, 1.5, 4)$ in the $(x_1, x_2, h_{11}, h_{12}, h_{22}, y)$-space.

We now perform the main step of `FastC2V`, which is to swap upper bounding inequalities based on $p$. They are swapped to whichever inequality is violated by $p$, or not swapped if no inequality is violated for a given ReLU neuron.

In this example, we skip $h_{11}$ and $h_{12}$ for simplicity as no swapping occurs, and we focus on $h_{22}$. For $h_{22}$, the relevant values of $\bar{p}$ are $\bar{h}_{11} = 1$, $\bar{h}_{12} = 1.5$, and $\bar{h}_{22} = 1.5$. Normally, we would solve the separation problem at this point, but for illustrative purposes we list out all possible upper bounding inequalities that we can swap to.

Recall Theorem 1 and compute:

$$\ell(\varnothing) = 2 \qquad\qquad \ell(\{1\}) = -2.5$$
$$\ell(\{2\}) = 0.5 \qquad\qquad \ell(\{1,2\}) = -4$$

(a) Original inequality from the $\Delta$-relaxation.

(b) Inequality (31).

(c) Inequality (32).

Figure 5: Three options of upper bounding inequalities for $h_{22}$. The black point depicts the solution that we would like to separate, which is cut off by inequality (31).

Based on these values, we have $\mathcal{J} = \{(\varnothing, 1), (\{2\}, 1)\}$, or in other words, two possible inequalities to swap to. By following the formulation in Theorem 1, we obtain the inequalities

$$h_{22} \leqslant -\frac{2}{3}h_{11} + 2 \tag{31}$$

$$h_{22} \leqslant -\frac{1}{6}h_{11} + h_{12} + \frac{1}{2} \tag{32}$$

These inequalities are illustrated in Figure 5. We observe that our point $p$ is cut off by the inequality (31): $1.5 = \bar{h}_{22} > -\frac{2}{3}\bar{h}_{11} + 2 = \frac{4}{3} \approx 1.333$. Therefore, for this neuron, we swap the upper bounding inequality to (31). In other words, our pair of inequalities for $h_{22}$ is now:

$$0 \leqslant h_{22} \leqslant -\frac{2}{3}h_{11} + 2$$

The last step of `FastC2V` is to redo the backward propagation with the swapped inequalities and recompute the bound. We obtain:

$$\begin{aligned}
y = h_{12} + h_{22} + 1 &\leqslant h_{12} + \left(-\frac{2}{3}h_{11} + 2\right) + 1 \\
&= -\frac{2}{3}h_{11} + h_{12} + 3 \\
&\leqslant -\frac{2}{3}(-x_1 + x_2 + 1) + \left(-\frac{3}{4}x_1 + \frac{3}{4}\right) + 3 \\
&= -\frac{1}{12}x_1 - \frac{2}{3}x_2 + \frac{37}{12}
\end{aligned}$$

Solving $\max_{x \in [-1,1]^2} -\frac{1}{12}x_1 - \frac{2}{3}x_2 + \frac{37}{12}$ gives us an improved bound of $\frac{23}{6} \approx 3.833$, completing the `FastC2V` algorithm for upper bounding $y$. Note that this procedure is not guaranteed to improve the initial bound, and in general we take the best between the initial bound and the new one.

Incidentally, we observe in Figure 5(a) that the big-$M$ inequality from the $\Delta$-relaxation is, in general, not facet-defining for the convex hull of the feasible points depicted in blue. This explains why it can not be directly reconstructed from our convex hull description (6).

## E  Implementation details

In this section, we add to the implementation details provided in Section 5.

The implementation of the propagation-based algorithm involves the following details:

- It may occur that the result of Algorithm 1 has zero coefficients for some variables $x_i$, in which case any feasible value for $x_i$ produces an optimal solution. For those variables, we select the midpoint between the lower bound and upper bound to proceed with Algorithm 2.

- We find that running more than one iteration of the propagation-based algorithm does not yield improving results. A possible reason for this is that while these inequalities are stronger in some portions of the input space, they are looser by themselves in others, and balancing this can be difficult. Improving this trade-off however is outside the scope of this paper.

- We use no tolerance on violation. That is, every violated inequality is swapped in.

The implementation of the LP-based algorithm involves the following details:

- We find that the Conv networks examined are very numerically unstable for LPs due to the presence of very small weights in the networks. Taking no action results in imprecise solutions, sometimes resulting in infeasible LPs being constructed. To improve on this instability, we consider as zero any weight or generated bound below $10^{-5}$. In addition, we run `DeepPoly` before the LP to quickly check if the neuron can be linearized. This is applied only to the LP-based methods. Note that the default feasibility and optimality tolerances in Gurobi are $10^{-6}$. With this, we end up solving an approximate problem rather than the exact problem, though arguably it is too difficult to solve these numerically unstable LPs with high precision and reasonable time in practice.

- For separation, we implement the $O(n \log n)$ version of the algorithm based on sorting instead of the $O(n)$ version.

- For each bound computed, we generate new cuts from scratch. More specifically, when solving for each bound, we make a copy of the model and its LP basis from the previous solve, run the LP solve and cut loop, retrieve the bound, and then discard this copy of the model.

- We add cuts whose violation exceeds a tolerance of $10^{-5}$.

- In the context of mixed-integer programming, it is well known that selecting a smaller subset of cuts to add can be very beneficial to reduce solving time, but for simplicity, we perform no cut selection in this method.

- An alternative to the LP-based method is to solve a MIP with analogous cutting planes with binary variables [3], but we find that this method, free of binary variables, is more lightweight and effective even without cut selection and all the presolve functionalities of modern MIP solvers. The ability to solve these LPs very quickly is important since we solve them at every neuron. In addition, this gives us more fine-grained control on the cuts, providing a better opportunity to evaluate our inequalities.

The implementation of all algorithms involve the following details:

- We attempt to linearize each neuron with simple interval arithmetic before running a more expensive procedure. This makes a particularly large difference in solving time for the Conv networks, in which many neurons are linearizable.

- As done in other algorithms in the literature, we elide the last affine layer, a step that is naturally incorporated in the framework from Section 4.1. In other words, we do not consider the last affine layer to be a neuron but to be the objective function.

- We fully compute the bounds of all neurons in the network, including differences of logits. We make no attempt to stop early even if we have the opportunity to infer robustness earlier.

- When solving the verification problem, scalar bounds on the intermediate neurons only need to be computed once per input image (i.e. once per set $X$), and can be reused for each target class (i.e. reused for different objectives $c$).

The details of the networks from the ERAN dataset [38] are the following. To simplify notation, we denote a dense layer by `Dense(size, activation)` and a convolutional layer by `Conv2D(number of filters, kernel size, strides, padding, activation)`.

- 6x100: $5\times$ `Dense(100, ReLU)` followed by `Dense(10, ReLU)`. This totals 510 units. Trained on the MNIST dataset with no adversarial training.

- 9x100: $8\times$ `Dense(100, ReLU)` followed by `Dense(10, ReLU)`. This totals 810 units. Trained on the MNIST dataset with no adversarial training.

Figure 6: Number of verified images by each method given various values of the allowed distance from the base image. Lines are averages over 16 randomly initialized networks and error bands represent standard deviation.

- 6x200: $5 \times$ `Dense(200, ReLU)` followed by `Dense(10, ReLU)`. This totals 1010 units. Trained on the MNIST dataset with no adversarial training.

- 6x200: $8 \times$ `Dense(200, ReLU)` followed by `Dense(10, ReLU)`. This totals 1610 units. Trained on the MNIST dataset with no adversarial training.

- MNIST ConvSmall: `Conv2D(16, (4,4), (2,2), valid, ReLU)`, `Conv2D(32, (4,4), (2,2), valid, ReLU)`, `Dense(100, ReLU)`, `Dense(10, linear)`. This totals 3604 units. Trained on the MNIST dataset with no adversarial training.

- MNIST ConvBig: `Conv2D(32, (3,3), (1,1), same, ReLU)`, `Conv2D(32, (4,4), (2,2), same, ReLU)`, `Conv2D(64, (3,3), (1,1), same, ReLU)`, `Conv2D(64, (4,4), (2,2), same, ReLU)`, `Dense(512, ReLU)`, `Dense(512, ReLU)`, `Dense(10, linear)`. This totals 48064 units. Trained on the MNIST dataset with DiffAI for adversarial training.

- CIFAR-10 ConvSmall: `Conv2D(16, (4,4), (2,2), valid, ReLU)`, `Conv2D(32, (4,4), (2,2), valid, ReLU)`, `Dense(100, ReLU)`, `Dense(10, linear)`. This totals 4852 units. Trained on the CIFAR-10 dataset with projected gradient descent for adversarial training.

# F   Supplementary computational results

We computationally examine the sensitivity of the algorithms in this paper to different training initializations and distances from the base image.

We focus on networks for the MNIST dataset. The first two architectures 6x100 and 6x200 have 6 hidden layers of 100 and 200 ReLUs respectively, followed by a linear output layer of 10 ReLUs (this differs slightly from the ERAN networks of the same name described in Appendix E). The MNIST ConvSmall architecture is the same as described in Appendix E. Average test accuracies are 97.04%, 97.61%, and 98.63% respectively. For each architecture and distance, we train 16 randomly initialized networks. Each network is trained with a learning rate of 0.001 for 10 epochs using the Adam training algorithm, without any adversarial training.

Figure 6 illustrates the average number of verified images. The error bands represent standard deviation over the 16 networks. We observe that OptC2V and FastC2V perform well across different networks and distances.

In addition, Figure 7 depicts survival plots for the results from Table 1: the number of images that can be verified given individual time budgets.

Figure 7: Survival plots for the results in Section 5. The horizontal dashed line is the upper bound on the number of verifiable images.