[Reviews · NeurIPS 2020]

Review 1

Summary and Contributions: In a nutshell, this paper reads like the algorithm-focused companion of the paper "Strong mixed-integer programming formulations for trained neural networks". The paper describes a tight convex relaxation of the multivariate inputs to a single ReLU (as opposed to the now standard triangle-relaxation). Even thought the tight description has exponentially many facets, performing separation over this set is easy. This separation algorithm is then utilized to build two new algorithms for verification -- (1) a cutting plane approach over the LP relaxation, (2) an approach that iteratively tightens bounds by doing forward/backward passes over the network. Empirical results over small-ish networks demonstrate a significant improvement in performance over prior methods.

Strengths: This paper is a good example of using mathematical intuition + mathematical theory to bring about new _practical_ algorithms for a problem that has seen a lot of attention in recent years. The paper brings tools from discrete optimization to bear on the problem of performing robustness verification on feedforward neural networks with ReLU neurons. The high-level ideas are clean and easy to understand for those with a background from that community. There is a good amount of theory and explanations that could be of interest to the community. The experimental results are solid.

Weaknesses: This paper is very dense and can be hard to read for some. See my comments under "Clarity".

Correctness: To my knowledge, yes.

Clarity: It is well-written but dense. The presentation of the paper is similar to what you might see in Mathematical Programming but crammed into the NeurIPS format, with some additional explanations to make the work appeal to a wider ML audience. I am familiar with works in the MP/IPCO community and so I found it reasonable to follow, but sometimes I got lost in the notation. I can imagine that it could be harder for some others. If the other reviewers concur with this sentiment, I would suggest the following changes to better appeal to the NeurIPS audience: - Defer more of the math and propositions/corollaries to the supplementary material. I understand the desire to get all the statements in the main paper to showcase your theoretical contributions, but I think this obfuscates the practical algorithmic contributions of the paper. - Add a figure to illustrate the basic high-level ideas behind the algorithms in the main text. This figure would also be useful for the eventual poster/presentation of this work. It might also be the case that giving the main text more room to breathe (e.g. an arxiv version with more pages) might suffice.

Relation to Prior Work: I would like a little discussion of how this work builds on or differs from the aforementioned prior work. The authors can decide if it's appropriate to do so in the paper itself or just in the response.

Reproducibility: Yes

Additional Feedback: As stated earlier, the main issue is the presentation. Another thing that would be nice is providing performance profiles for the experiments in the supplementary. Do the authors know how this work empirically compares with "Neural Network Branching for Neural Network Verification" by Jingyue Lu, M. Pawan Kumar? I would love to see the code released. ---------------------------------- Update after rebuttal: I would like to thank the authors for their detailed response. I have increased my score to an 8 and I look forward to reading a version of this paper with better presentation.


Review 2

Summary and Contributions: ***Update after author feedback*** Authors have addressed some of my concerns in feedback. After some reflection, I tend to *conditionally* increase my score (from 6 to 7) because of the following reasons: 1. they have provided partial evidence of assessment of variance in the experiments, showing that results are significant. It is crucial that such variance estimates are added to the paper and mentioned. There are strong arguments from a majority in the community, that not assessing variance arising from randomness is an experimental flaw. This is the *conditional* part of my score increase 2. The fact that they provide a separation oracle for the *tightest* convex relaxation of ReLU, helps to close down the chapter on such type of methods based on convex relaxations of activations. I find this novelty significant. *** Original Review *** The paper introduces a tighter convex relaxation, than the triangle-relaxation, for the problem of certifying ReLU networks i.e., certifying that all elements in a certain set are assigned the same label. The triangle relaxation has been the basis of many previous certification methods. The set defined in the new convex relaxation is shown to admit an efficient separation oracle, and thus the optimization problem is computationally tractable (polynomial time). Empirically there is some improvements over the baselines, but with larger computational budget.

Strengths: THe results are strongly rooted in theory, in particular, proving the existence of an efficient separation oracle shows that the optimization problem is computationally tractable. The empirical evaluation is also done on what appears to be a standard benchmark (ERAN dataset) and there is some improvement over the baselines.

Weaknesses: The work lacks novelty, as the certification problem has been extensively studied. In my opinion, it is not clear how much more can it be improved, and how relevant is in applications. Also, the dataset proposed (ERAN) does not have multiple trained versions of the same network. Given that such networks are trained using stochastic methods like SGD, there might potentially be a high variance in the numerical results. So, it is hard to convince the reader that the empirical improvement is significant. I would suggest to run the certification procedure on multiple versions of the same network (trained with different seed), to obtain some measure of variance in the numerical results.

Correctness: The supplementary seems complete and supporting of the main claims. I have not checked thoroughly, as it is quite long. The empirical evaluation is done on a standard dataset. However such dataset lacks repeated runs of the same network (with different seeds), and thus there is no measure of variance in the numerical results.

Clarity: The paper is clear and it is not hard to follow.

Relation to Prior Work: the work compares to multiple baselines on a standard dataset. The main differences are discussed and clear from the main text.

Reproducibility: Yes

Additional Feedback:


Review 3

Summary and Contributions: The authors proposed a explicit linear inqeuality description for the tightest possible convex relaxation of a single neuron. They show that there's an efficient separation routine and present a linear time algorithm to evaluate point.

Strengths: clear presentation for the algorithm and proofs.

Weaknesses: similar performance compared to Kpoly

Correctness: yes

Clarity: yes

Relation to Prior Work: yes

Reproducibility: No

Additional Feedback:

[Author Response · NeurIPS 2020]

We appreciate the reviewers' feedback. It is encouraging that the reviewers highlighted how the work is effectively able to develop mathematical theory into practical verification algorithms (R1, R2) and the overall writing quality (R1, R2, R3). The main concerns were on presentation to a non-optimization audience (R1), clarifications on how it relates to specific works (R1), and potential variance in numerical results (R2). We address all of these concerns below and also make clarifications regarding comments on novelty (R2) and computational performance (R3).

[R1] **"The high-level ideas are clean and easy to understand for those with a background from [the discrete optimization] community... This paper is very dense and can be hard to read for some."** It is encouraging to hear this positive comment from R1 and we understand the concern regarding a different audience (although all other reviewers made positive comments on clarity). This was kept top of mind during writing but striking this balance is difficult given that we value mathematical rigor. To improve this, we will add a new figure as R1 suggested and revisit the writing, although we prefer to keep most of the math in the main text as we believe they are important contributions.

[R1] **"... paper reads like the algorithm-focused companion of [1] ", "I would like a little discussion of how this work builds on or differs from [1]".** We approached this in the footnote of p.4 and Appendix A (albeit briefly), but the final version will contain an expanded discussion that further clarifies how the manuscript differs in crucial ways from [1]. The central difference is that [1] works in the MIP space (i.e. with binary variables) whereas we work in the original space, with the connection that our formulation is a projection of the LP relaxation in [1]. Our main theoretical result is a minimal description of the new formulation, which can only be attained with a careful and data-dependent analysis of the structure of the ReLU (standard projection methods would lead to large, redundant formulations). This also requires the development of a more involved separation algorithm. Algorithmically, by working without binary variables, we are able to develop our fast propagation-based algorithm FastC2V, and no similar algorithm exists in [1].

[R1] **"how this work ... compares with [the work] by Lu and Pawan Kumar?"** Thanks, we were not aware of this work and we will cite it in the introduction. A detailed comparison is beyond the scope of our work given that we focus on relaxed verifiers (i.e. bounds) rather than branching-based exact verifiers. Nonetheless, our work could be used to directly improve theirs. First, exact verifiers typically use fast relaxed verifiers as a subroutine to produce bounds, and FastC2V can play that role. Second, our inequalities can improve the relaxation used in branching-based methods.

[R1] **"Another thing that would be nice is providing performance profiles ..."** We will include cactus plots (number of images verified per time) in the appendix of the final version (we slightly prefer these over performance profiles).

[R2] **"... there might potentially be a high variance in the numerical results."** We will add an additional section in the appendix investigating variance for networks trained with different initializations. We intend to keep the main experiments as is, so that we can keep the comparison with previous algorithms, and train networks ourselves for these supplementary experiments. As a tiny partial preview, we compared DeepPoly vs our FastC2V for 100 images in a 6x100 network in 10 adversarially trained networks with different random initializations (implementation and parameters are different from those in main text; we may adjust these further), and in all networks our FastC2V verifies more images than DeepPoly, averaging 49.0 vs 67.2 images verified, with standard deviations 3.41 and 4.09 respectively.

[R2] **"The work lacks novelty, as the certification problem has been extensively studied. In my opinion, it is not clear how much more can it be improved, and how relevant is in applications."** We strongly believe that the technical content of the paper is novel within the area of certification (at the very least). We provide for the first time the tightest possible convex relaxation for a ReLU neuron (without binary variables) and a new fast propagation-based algorithm that effectively leverages this new formulation, backed by computational evidence. We push the convex relaxation barrier from [2] and improve upon the kPoly method by [3] (both published in NeurIPS 2019). We believe that the steady abundance of work in the area of certification serves as evidence that it is active and relevant.

[R3] **"Similar performance compared to kPoly".** We would like to clarify that our computational contribution is much broader than this characterization. We highlight that FastC2V can verify more images than the strongest possible convex relaxation defined in [2] (i.e. solving the "triangle relaxation" LP) with ~5-50x faster solve time in our instances. As discussed in [2], this "triangle relaxation" is a barrier that restricts the great majority of verification algorithms. Even if one considers kPoly [3], a state-of-the-art algorithm that does bypass this barrier, we obtain better verification capability with our OptC2V. Finally, mathematical foundation aside, the algorithms themselves are simpler to implement (the pseudocode for FastC2V given in the appendix fits in a few pages) and depend less on hyperparameters than kPoly.

[R3] **Reproducibility.** R3 responded "no" to reproducibility whereas R1, R2 responded "yes". The algorithms are described in the main text and trained networks are publicly available, but please also refer to the appendix for pseudocode and implementation details. Furthermore, we plan to open source the code (as suggested by R1).

**References.** [1] Anderson et al., "Strong mixed-integer programming formulations for trained neural networks". 2020.
[2] Salman et al., "A convex relaxation barrier to tight robustness verification of neural networks". In NeurIPS 2019.
[3] Singh et al., "Beyond the single neuron convex barrier for neural network certification". In NeurIPS 2019.

[Meta-Review · NeurIPS 2020]

This paper provides a clever method for tightening a well-known convex relaxation.